# On the Inadequacy of Similarity-based Privacy Metrics: Reconstruction Attacks against "Truly Anonymous Synthetic Data"

## Abstract

Training generative models to produce synthetic data is meant to provide a privacy-friendly approach to data release. However, we get robust guarantees only when models are trained to satisfy Differential Privacy (DP). Alas, this is not the standard in industry as many companies use ad-hoc strategies to empirically evaluate privacy based on the statistical *similarity* between synthetic and real data.

In this paper, we review the privacy metrics offered by leading companies in this space and shed light on a few critical flaws in reasoning about privacy entirely via empirical evaluations. We analyze the undesirable properties of the **most popular** metrics and filters and demonstrate their unreliability and inconsistency through counter-examples. We then present a reconstruction attack, *ReconSyn*, which successfully recovers (i.e., leaks all attributes of) at least 78% of the low-density train records (or outliers) with only black-box access to a single fitted generative model and the privacy metrics. Finally, we show that applying DP **only to the model** or using low-utility generators does not mitigate *ReconSyn* as the privacy leakage **predominantly** comes from the metrics. Overall, our work serves as a warning to practitioners not to deviate from established privacy-preserving mechanisms.

## 1 Introduction

Synthetic data – i.e., artificially generated data produced by machine learning algorithms – has attracted growing interest not only from the research community (Jordon et al., 2022), but also regulatory bodies (Information Commissioner's Office, 2022; Financial Conduct Authority, 2023), non-profits (UN, 2023; OECD, 2023), and government agencies (Benedetto et al., 2018; NIST, 2018; 2020). It promises a drop-in replacement for sensitive data in various use cases, e.g., private data release, de-biasing, augmentation, etc. Numerous providers of synthetic data solutions have entered a flourishing market attracting considerable investments (Crunchbase, 2022; TechCrunch, 2022; Forbes, 2022), with products serving large corporations in various sectors.

The basic idea behind synthetic data is to rely on generative machine learning models, learning the probability distribution of the real data and creating new (synthetic) records by sampling from the trained model. However, models trained without robust privacy guarantees can overfit and memorize individual data points (Carlini et al., 2019b; Webster et al., 2019), which enables attacks like membership and property inference (Hayes et al., 2019; Hilprecht et al., 2019; Chen et al., 2020; Stadler et al., 2022; Annamalai et al., 2023). This, in turn, could lead to disastrous breaches and leakage of individuals' health, financial, and other sensitive data.

**Main Motivation.** The established framework to bound information leakage and defend against privacy attacks is Differential Privacy (DP) (Dwork et al., 2006; Dwork & Roth, 2014). Specifically, for synthetic data, one needs to train generative models while satisfying DP (Zhang et al., 2017; Jordon et al., 2018; McKenna et al., 2021). While almost all companies in this space claim their synthetic data products meet regulatory requirements such as GDPR, HIPAA, or CCPA, we find that they rarely use DP, as shown in App. A. This is worrisome, as models are often trained on sensitive data in highly regulated environments (e.g., medical applications (Hradec et al., 2022)).

Rather than relying on well-established privacy notions, many companies use ad-hoc heuristics to guarantee privacy empirically; see, e.g., (Platzer & Reutterer, 2021; Mobey Forum, 2022). Some combine **unperturbed** heuristics with DP, **breaking the end-to-end DP pipeline**, which ultimately negates its privacy protections, as our evaluation will demonstrate. In fact, even research papers, e.g., in the medical domain, have proposed models that exclusively rely on similar empirical privacy heuristics (Park et al., 2018; Lu et al., 2019; Yale et al., 2019; Zhao et al., 2021; Guillaudeux et al., 2023; Liu et al., 2023; Yoon et al., 2023).

**Problem Statement.** The heuristics used in industry mainly consist of privacy metrics and filters based on similarity (see Sec. 2), i.e., how close synthetic records are to their nearest neighbor in the train data. If enough synthetic points are too close, according to pre-configured statistical tests vs. holdout test data, they are filtered out, or the whole sample is discarded; otherwise, the data is considered safe. The idea is that synthetic data should be similar and representative of the train data but not too close, which intuitively makes sense. However, the meaningfulness of these heuristics has not been rigorously studied. This motivates assessing the validity of entirely relying on empirical evaluation based on distances and whether this approach risks providing false protection claims.

**Technical Roadmap.** We explore, characterize, and analyze the major disadvantages of the most commonly used privacy metrics/filters in industry. (In App. E, we also show counter-examples whereby, even if all privacy tests pass, privacy violations and inconsistencies can still occur.)

**Then, we propose *ReconSyn*, a proof of concept black-box attack designed to highlight the inherent weaknesses of the privacy metrics in synthetic data generation.** The attack recovers train data from low-density regions (where the most at-risk records reside) with realistic assumptions. **Besides the privacy metrics, the adversary can only access a single fitted generative model.** In fact, the attack is agnostic to the generative approach, the type of dataset, and use case, etc.

**Experimental Evaluation.** In addition to the counter-examples, we present experiments demonstrating the effectiveness of *ReconSys* vis-à-vis five state-of-the-art **tabular** models (PrivBayes, MST, DPGAN, PATE-GAN, CTGAN) and five commonly used datasets, including Adult, Census, and MNIST. The attack reconstructs at least 78% of the underrepresented train data records (or outliers) with perfect precision in all settings **(see Fig. 1)**. Some models are more vulnerable: attacking graphical models (PrivBayes, MST) requires fewer rounds to achieve similar results than GANs. In fact, *ReconSyn* is successful even when attacking low-utility generators such as Random and Independent.

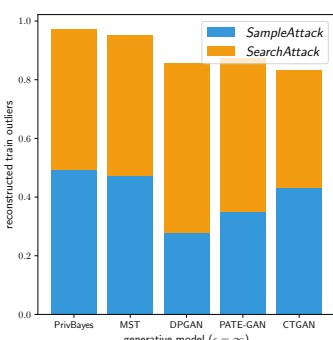

**Figure 1:** *ReconSyn* perfomance.

**Main Contributions:**

1. We are the first to analyze the undesirable properties of the most common privacy metrics and filters used in industry to empirically "guarantee" the privacy of synthetic data.
2. We propose a novel reconstruction attack, *ReconSyn*, with minimal and realistic assumptions – black-box access to a single trained generative model and the privacy metrics.
3. We demonstrate that applying DP **to the generative model** does not mitigate *ReconSyn* when combined with **unperturbed** heuristics, as leakage persists through the metrics.
4. We show that using similarity-based privacy metrics does not provide GDPR compliance.
5. We discuss how, assuming a similar threat model, *ReconSyn* can be adapted to other attacks like membership and attribute inference.

Overall, our work prompts the need to move away from attempts to guarantee privacy in an ad-hoc, empirical way. We believe our findings will be useful to practitioners when deploying solutions requiring the processing of sensitive data, as well as policymakers when creating standards and best practices for privacy-preserving synthetic data adoption.

*NB:* We have shared our work with the relevant synthetic data companies in the spirit of responsible disclosure and are working with them on the next steps – see Ethics Statement.

## 2 PRIVACY METRICS AND FILTERS

In this section, we present the three privacy metrics and two filters broadly used by synthetic data companies to guarantee privacy (see App. A). The former are used to measure the privacy of the synthetic data and run pass/fail statistical tests, while the latter remove records from the generated data points based on their similarity to train records or outliers.

A common implementation pre-processing step, which, unless stated otherwise, we also follow, is to *discretize* the data. The input to all metrics is train, synthetic, and holdout test dataset, $D_{test}^n$ (with the same size, $n$, as the train data, $D_{train}^n$) – which comes from the same distribution as $D_{train}^n$) but was not used to train the generative model – to serve as a reference.

**Intuition.** The overall idea behind similarity-based privacy metrics is that synthetic records should be as close as possible to train ones, but not too close, i.e., not closer than what would be expected from the holdout records (Platzer & Reutterer, 2021). More precisely, we compute the closest pairwise distances (for discrete data, we use Hamming distance, for continuous – Euclidean) for $(D_{train}^n, D_{synth}^{n'})$ and $(D_{train}^n, D_{test}^n)$, and run a pass/fail test.

The passing criterion is a comparison between simple statistics calculated from the two distributions – the average or the 5th percentile, while the output of the metrics is the pass/fail flag alongside the actual statistics as per (MOSTLY AI, 2022b). If all three tests pass, the data becomes "truly anonymous synthetic data" (MOSTLY AI, 2020) and could be freely shared alongside the privacy scores. Otherwise, the synthetic data is not considered safe enough to be released.

**Identical Match Share (IMS)** is a privacy metric that captures the proportion of identical copies between train and synthetic records. The test passes if that proportion is smaller or equal to the one between train and test datasets. In practice, IMS is used or advocated by (MOSTLY AI, 2020; Syntegra, 2021; DataCebo, 2022), and others (Lu et al., 2019; ONS DSC, 2022; AWS, 2022).

**Distance to Closest Records (DCR)** also compares the two sets of distances. It looks at the overall distribution of the distances to the nearest neighbors or closest records. The test passes if $(D_{train}^n, D_{synth}^{n'})$-5th percentile is larger or equal **than the other pair**. DCR is supposed to protect against settings where the train data was just slightly perturbed or noised and presented as synthetic. Tonic (2023a); MOSTLY AI (2020); Hazy (2023b); Syntegra (2021); Statice (2023a), and several scientific studies/blogposts (Park et al., 2018; Lu et al., 2019; Yale et al., 2019; Zhao et al., 2021; ONS DSC, 2022; AWS, 2022; Guillaudeux et al., 2023; Liu et al., 2023; Yoon et al., 2023) use DCR.

**Nearest Neighbor Distance Ratio (NNDR)** is very similar to DCR, but the nearest neighbors' distances are divided by the distance to the second nearest neighbor. The idea is to add further protection for outliers by computing relative rather than absolute distances. NNDR, too, compares the 5th percentile between the two sets of distributions, and is used by (MOSTLY AI, 2020) and in academic papers (Zhao et al., 2021; Guillaudeux et al., 2023).

**Similarity Filter (SF)** is similar in spirit to the privacy metrics, but rather than just measuring similarity, it excludes or filters out individual synthetic data points if they are identical or too close to train ones. Essentially, SF aims to ensure that no synthetic record is overly similar to a train one. It is used by (Replica Analytics, 2020; Gretel, 2021; Synthesized, 2023a).

**Outlier Filter (OF)** focuses on the outliers; it removes synthetic records that could be considered outliers with respect to the train data, and is used by (Gretel, 2021).

**Passing Criteria.** Throughout our experiments, we adopt the criteria from (MOSTLY AI, 2020), unless stated otherwise, i.e., a synthetic dataset (for whose generation none of the filters were used) is considered private if all three privacy tests—**coming from IMS, DCR, and NNDR**—pass.

**Additional Background Information.** We defer the related work on reconstruction in databases and machine learning to App. B. In App. C, we outline common notation on synthetic data and details about the generative models, **datasets, and the criteria for defining outliers** in our evaluation.

## 3    Fundamental Limitations of Similarity-Based Privacy Metrics

In this section, we identify and discuss several issues with using similarity-based privacy metrics (SBPMs) to guarantee privacy through pass/fail tests. We later exploit these properties to build a successful reconstruction attack.

***Issue 1: No Theoretical Guarantees.*** First and foremost, SBPMs do not provide any theoretical or analytical guarantees. They do not define a threat model or a strategic adversary, thus ignoring some of the most fundamental security principles (Anderson, 2020). Instead, SBPMs rely on a number of arbitrarily chosen statistical tests.

This prompts a few questions, e.g., why choose these specific tests instead of others? What exactly do they protect against? How were the passing criteria selected? Furthermore, SBPMs do not rule out vulnerabilities to current or future adversarial attacks, including *ReconSyn* (see Sec. 4).

***Issue 2: Privacy as Binary Property.*** SBPMs treat privacy leakage as a binary property, i.e., the synthetic data is either "truly" private or not. This is despite the fact that SBPMs do not rely on, e.g., an adversarial advantage that can be proven asymptotically small under certain assumptions, e.g., as done in Cryptography. In fact, using pass/fail tests removes analysts' sense of direction and ability to measure privacy leakage across a continuous interval.

This has two consequences. First, it is hard to know what choices (e.g., models, hyperparameters, etc.) contribute to making the synthetic data private. Second, releasing a single private synthetic dataset is deemed as safe as releasing many (as long as they pass the tests), even though this increases leakage since the provider needs to call the train/test data every time new data is generated. Arguably, this is related to the "Fundamental Law of Information Reconstruction" (Dwork & Roth, 2014), stating that overly accurate answers to too many questions will destroy privacy in a spectacular way.

***Issue 3: Non-Contrastive Process.*** SBPMs are computed in a non-contrastive way. That is, they do not compare the computations when an individual is included or not. Since there is no noise or randomness ingested into the process, plausible deniability is ruled out. Thus, calculating the privacy metrics leads to a variety of attacks, including simple ones like differencing attacks. For example, if an adversary makes two calls to the metrics, one with and one without a particular individual, they can deduce some information (e.g., whether the individual is an exact match or closer than 5th percentile) with 100% confidence since the computations will carry no uncertainty.

***Issue 4: Lack of Worst-Case Analysis.*** All SBPMs use simple statistics (average or 5th percentile) as passing criteria. This leaves room for maliciously crafted synthetic datasets that might pass the tests but still reveal sensitive data. Also, this does not protect against worst-case scenarios, i.e., memorization and replication of outliers, which, combined with the lack of plausible deniability (from Issue 3), increases the adversary's chance of launching a successful attack.

Unfortunately, using a held-out dataset for comparison does not alleviate the problem due to what is commonly and informally defined as the "Generalization Implies Privacy" fallacy, i.e., privacy is a worst-case problem while generalization is an average-case. Put simply, even if all tests pass, i.e., the model generalizes, memorization cannot be ruled out (Song et al., 2017).

***Issue 5: Privacy as Data Property.*** SBPMs expect a single synthetic dataset as input, which has several implications. First, it means we measure the privacy of a specific dataset and not the generative model/process. Therefore, privacy becomes a property of the data rather than the generating process. Also, SBPMs require running the metrics on each and every generated synthetic data in order to guarantee privacy which, unfortunately, actually leaks more privacy (as discussed in Issue 2). Second, the specific synthetic dataset may or may not be representative of the distribution captured by the model, which could lead to inconsistent results across generation runs. Typically, privacy is defined as a statistical property over many such instances.

Due to space limitation, we present the remaining **three** limitations in App. D – incorrect interpretation, risk underestimation, and implementation challenges. We also illustrate the inconsistency and untrustworthiness of the metrics through counter-examples in App. E.

**Take-Aways.** In summary, relying entirely on empirical evaluations to "guarantee" privacy present several critical weaknesses that may lead to an artificially high sense of security. Unfortunately, this approach is ineffective and embeds severe vulnerabilities to privacy attacks.

**Figure 2:** Overview of *ReconSyn*. The provider **1.** splits the real data into train/test, **2.** fits a generative model on the train data, **3.** generates synthetic data (privacy filters are applied), **4.** runs the privacy metrics on the synthetic data. The adversary can make API calls (they have black-box access) to the fitted generative model and privacy metrics. They **a.** generate synthetic datasets, **b.** run them through the privacy metrics to observe the pass/fail tests and scores (if tests pass), **c.** reconstruct underrepresented train records (outliers) through *SampleAttack* and *SearchAttack* (introduced in Algorithm 1).

## 4 THE *ReconSyn* RECONSTRUCTION ATTACK

We now introduce a novel attack, *ReconSyn*, aimed at recovering the outliers in the train data with minimal assumptions. An overview of the attack is reported in Fig. 2.

**Adversarial Model.** A *synthetic data provider* has access to train and test datasets ($D_{train}^n$ and $D_{test}^n$), trains a generative model ($G_{\bar{\theta}}(D_{train}^n)$), generates synthetic datasets ($D_{synth}^{n'}$), and deems them private if they pass all privacy tests (a combination of privacy metrics and/or filters).

We assume a strategic *reconstruction adversary* with black-box access to the trained generative model ($G_{\bar{\theta}}(D_{train}^n)$) and the privacy metrics. The adversary has the capability to sample from the trained model to generate synthetic datasets. They can add or remove data points to/from the synthetic data and make calls to the metrics APIs to observe the outcome of the tests and, in case all tests pass, the scores. Their goal is to reconstruct, or completely violate the privacy of, the train data outliers ($D_{train}^{out}$) by building a collection of synthetic datasets considered private by the provider.

**Algorithm Steps.** *ReconSyn* **(pseudocode in Algorithm 1 in App. F) comprises two subattacks: 1)** *SampleAttack***, which generates and evaluates samples drawn from the generative model, and 2)** *SearchAttack***, which strategically examines the history of records generated in the first phase. Next, we offer an overview of the strategies and present more details in App. F.**

**As a first step, the adversary uses** *OutliersLocator* **to identify regions with underrepresented records or outliers. This involves generating a large synthetic data sample, fitting a Gaussian Mixture model, and selecting the smallest isolated clusters.**

*SampleAttack* **follows a simple procedure. In each round, it generates synthetic data, then identifies potential outliers using** *OutliersLocator***. It removes data already examined in previous rounds, as recorded in its history. The attack then queries the metrics API to check for exact matches (if all tests pass) and adds all queried data to the history.**

**Informally, the idea behind** *SearchAttack* **involves selecting close records from the history and 'shaking' or 'fixing' them one column at a time until an exact match is found. For a specific record, two steps are taken. We first identify columns that have not yet been reconstructed using its neighboring dataset, which is a square matrix where each row differs in a single column value. We then iteratively test possible values for these columns, filtering out records through** *OutliersLocator* **and the history. Ultimately, this leads to another match.**

**Plausibility of the Attack.** *ReconSyn* relies on three realistic assumptions, in that the adversary can:

1. Generate an unlimited amount of synthetic datasets, which is one of the main selling points for adopting synthetic data (Gretel, 2023a; MOSTLY AI, 2022a; Hazy, 2022).
2. Add or remove records – data augmentation is a popular use case advertised by synthetic data companies (Tonic, 2022; Gretel, 2023a; MOSTLY AI, 2023c).
3. Access the privacy tests and scores for every generation run (if all tests pass); again, this is explicitly offered by the main companies (MOSTLY AI, 2022b; 2023a).

Note that the adversary *does not have any side knowledge*: no access to the train/test data or even possession of data from the same distribution, no background information of the used generative

| Model | 2d Gauss Sample | Adult Small Sample | Adult | | Census | | MNIST | |
|---|---|---|---|---|---|---|---|---|
| | | | Sample | Search | Sample | Search | Sample | Search |
| **Oracle** | **0.95** | | | | | | | |
| **PrivBayes** | | 1.00 | 0.44 | 0.95 | 0.54 | 0.98 | 0.00 | 0.99 |
| **MST** | | 1.00 | 0.05 | 0.90 | 0.84 | 0.99 | 0.00 | 0.97 |
| **DPGAN** | | 0.96 | 0.02 | 0.78 | 0.15 | 0.82 | 0.00 | 0.97 |
| **PATE-GAN** | | 1.00 | 0.02 | 0.81 | 0.37 | 0.83 | 0.00 | 0.97 |
| **CTGAN** | | 0.99 | 0.00 | 0.80 | 0.74 | 0.90 | 0.00 | 0.80 |

**Table 1:** Overview of the performance of the *ReconSyn* attack against different models ($\epsilon = \infty$) and datasets.

approach, model, hyperparameters, nor model updates or gradients. They are also agnostic to the dataset type and the specific use case/downstream task.

**Why Outliers.** Our motivation for targeting the underrepresented regions in the train data is their potential correspondence with the most vulnerable individuals. **They are inherently more difficult to model accurately, which makes their reconstruction more challenging (see Sec. G).** Furthermore. outliers are at a higher risk of being memorized by models (Feldman, 2020) and are more susceptible to membership inference attacks (Stadler et al., 2022). Regulators, such as Information Commissioner's Office (2022), have explicitly highlighted the increased sensitivity of outliers.

**Why Reconstruction.** We choose to build a reconstruction attack as this is one of the most powerful attacks – it exposes *all* (sensitive) attributes – thus unequivocally demonstrating the untrustworthiness of similarity-based approaches to reason about privacy. If the attack is successful in reconstructing even a handful of train outliers with high precision, this will constitute a serious privacy violation (Carlini et al., 2022).

In fact, reconstruction implies the ability to single individuals out and enable their identification or link them to the real data. This, in turn, means that the process of generating synthetic data and guaranteeing its privacy has failed at least two of the three privacy guarantees outlined by European Commission Article 29 Working Party (A29WP, 2014), namely, singling out and linkability. Therefore, the process cannot be considered anonymous as per GDPR.

**Take-Aways.** *ReconSyn* is powerful and generalizable since it achieves both high recall and precision (see Sec. 5.1). Precision is perfect as we reconstruct outliers with 100% confidence (i.e., there are no false positives). Furthermore, assuming the same setup, other attacks such as membership and attribute inference could be considered specific subcases of *ReconSyn* (see App. H).

## 5 EVALUATION

In this section, we demonstrate that *ReconSyn* successfully recovers the train outliers in different settings. Our experiments are conducted against the models and datasets reviewed in App. C.

### 5.1 RECONSTRUCTION OF TRAIN OUTLIERS

We measure the performance of *ReconSyn* (*SampleAttack* and *SearchAttack*) against PrivBayes, MST, DPGAN, PATE-GAN, and CTGAN on increasingly more complex datasets (*2d Gauss*, *Adult Small*, *Adult*, *Census*, *MNIST*). Our experiments are summarized in Table 1. Since *ReconSyn* is highly successful in all settings, we do not report the utility of the generated synthetic data.

#### 5.1.1 *ReconSyn*, *SampleAttack*

We launch *SampleAttack* on all five datasets. We run it for 1,000 rounds for the first three datasets and for 5,000 rounds for the last two. The attack exhibits mixed results: regardless of the target model, it is very successful on *2d Gauss* and *Adult Small*, reconstructing at least 95% of train outliers, but struggles for the remaining three (see Table 1).

*2d Gauss*. Starting with *2d Gauss*, we attack the oracle and display the results in Fig. 10. Even though i) no generative model has been exposed to train data, and ii) the oracle has no memory of the synthetic data it has generated, *SampleAttack* manages to perfectly reconstruct 95% train outliers due to the privacy metrics' leakage. In other words, if the adversary had no access to the metrics, they would not be able to gain information about the train data by generating new data.

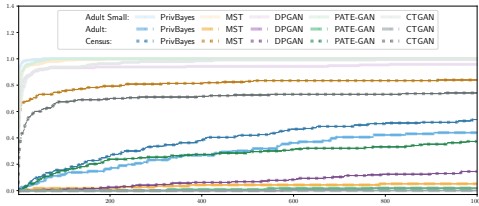

**Figure 3:** Proportion of reconstructed train outliers for increasing rounds by *SampleAttack*, *Adult Small*, *Adult*, and *Census*.

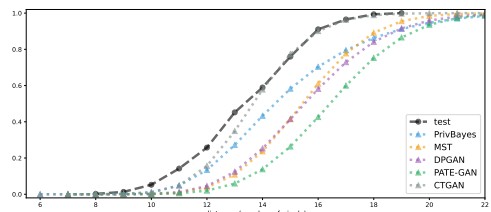

**Figure 4:** CDF of similarity distances between train and test/synthetic outliers by *SampleAttack*, *MNIST*.

***Adult Small.*** For *Adult Small*, we use *SampleAttack* against all five generative models and report the number of reconstructed outliers in Fig. 3 (top five lines). For PrivBayes, MST, and PATE-GAN, the attack quickly reconstructs around 90% outliers after just 10 rounds and eventually reaches 100%. For DPGAN and CTGAN, the attack plateaus at around 85% after 40 rounds, but by round 1,000, it slowly improves and achieves 96% and 99%, respectively. We believe that *SampleAttack* is extremely successful on this dataset because its domain is relatively small ($10^5$).

***Adult.*** On *Adult*, which has twice the number of columns and cardinality of $10^{15}$, the models are much less likely to memorize and reproduce individual data points. Indeed, apart from PrivBayes, *SampleAttack* only recovers 5% of the outliers – see Table 1 and Fig. 3 (bottom five lines).

***Census.*** Even though *Census* has roughly twice the columns/rows and much higher cardinality, *SampleAttack* is more successful (excluding PrivBayes on *Adult*), recovering on average 53% outliers (see middle lines in Fig. 3). Interestingly, attacking CTGAN yields better results than PrivBayes. Also, the recovery rate follows a linear trend (vs. logarithmic for *Adult Small*).

***MNIST.*** Finally, looking at *MNIST* which has even higher dimensionality/cardinality, *SampleAttack* fails completely and does not reconstruct even a single outlier. In fact, in Fig. 4 we see that CTGAN and PrivBayes generate images with the highest similarity to the real ones but still at a Hamming distance of at least 6 (i.e., number of different pixels). All models, however, create outliers further away from the train data compared to the distances between test and train. This is a confirmation that all privacy tests pass and test/synthetic datasets do not contain copies of the train data.

### 5.1.2 *ReconSyn*, *SearchAttack*

We run the follow-up *SearchAttack* on all models where *SampleAttack* achieves less than 95% reconstruction success, i.e., we attack all five models on *Adult*, *Census*, and *MNIST*. We run *SearchAttack* for up to $1/4$ of the distances; in other words, we go through the history and try to "fix" at most 4 columns of *Adult*/*Census* and 16 of *MNIST*. While widening the search to broader distances could lead to better results, we put the efficiency of our attack to the test by limiting the computations. In all cases, we manage to successfully reconstruct over 78% of all train outliers; see Table 1.

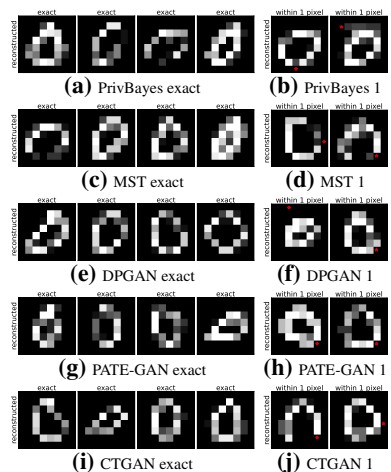

**Figure 5:** Reconstructed train outliers by *SearchAttack, MNIST.*

***Adult.*** For *Adult*, *SearchAttack* easily recovers the majority of train outliers, between 78%-95%. The attack is both more effective and efficient on the graphical models since it does not need to go back far in history—only a couple of distances (or columns). Most likely, this is due to: 1) *SampleAttack* was already more successful for these two models, generating more diverse data history, and 2) graphical models tend to overperform GANs on low-dimensional datasets like *Adult* and simple downstream tasks like marginal preservation (Ganev et al., 2023). Nonetheless, *SampleAttack* reconstructs most outliers against the GAN models, too, even though it requires searching further back (distance of 4).

***Census.*** *SearchAttack*'s performance on *Census* is similar – it reconstructs more outliers vs. the graphical models (99%) than the GANs (85% on average). Even though attacking PrivBayes starts at a disadvantage compared to CTGAN, *SearchAttack* manages to recover more outliers. As before, this could be because PrivBayes generates richer history and potential overfitting of CTGAN.

***MNIST.*** As for *MNIST*, despite the large data cardinality, *SearchAttack* reconstructs more than 80% of the train outliers. To reduce the search space, the adversary can be strategic, e.g., excluding some pixels (i.e., the ones on the sides of the image or the "frame") by setting their value to 0 after observing the common pattern after generating a collection of potential outliers. This way, the adversary is restricted and cannot fully reconstruct 21 out of the 488 outliers. Nonetheless, this could be considered a good trade-off vis-à-vis the number of saved computations – specifically, a factor of $\approx 480 = 30 \cdot 16$ (30 fixed pixels, 16 combinations per bin) per search. We report the number of exactly reconstructed train outliers and those within 1 pixel (even though the adversary can easily get an exact train data match by running the attack for one step without any restrictions). A subset of the recovered digits for all models is shown in Fig. 5.

Overall, *SearchAttack* is very successful, despite *SampleAttack*'s failure to recover any outliers – aside from CTGAN, attacking all other models results in reconstructing at least 97% of outliers. This might be because the generators manage to create diverse synthetic images not too dissimilar from the outliers (as already shown in Fig. 4). Conversely, even though CTGAN generates the closest images, that does not result in recovering more outliers. Potentially, this could be due to a mode collapse or the model's specific strategy of embedding categorical columns (both DPGAN and PATE-GAN use simple one-hot encoding). Unsurprisingly, out of the restricted 21 outliers, the adversary attacking CTGAN manages to recover only 6 compared to at least 16 for the other models.

### 5.1.3 TAKE-AWAYS

Our novel attack, *ReconSyn*, successfully reconstructs at least 78% of the train outliers with all tested models and datasets. *SearchAttack* performs better on lower-dimensional datasets but fails to recover any records for *MNIST*. However, the follow-up *SearchAttack* achieves an average of 90% success on the wider datasets and is slightly more successful when launched against graphical models.

### 5.2 DP AND LOW UTILITY GENERATIVE MODELS

#### 5.2.1 DP GENERATIVE MODELS

We now assess whether **training the generative models with DP guarantees** can prevent or minimize the performance of *ReconSyn*. **We simulate company product deployments which combine DP training with unrestricted metric access to the train data (see App. A).** We experiment with the 4 models relying on different mechanisms – namely, Laplace, Gaussian, DP-SGD, and PATE – while varying the privacy budget in the range $\{\infty, 1, 0.1\}$ on *Adult Small*. We keep $\delta$ constant to $1/n$. Again, we launch *SampleAttack* (1,000 rounds) on all models and *SearchAttack* (up to 1 column) in the cases where the former fails to achieve at least 95% reconstruction success.

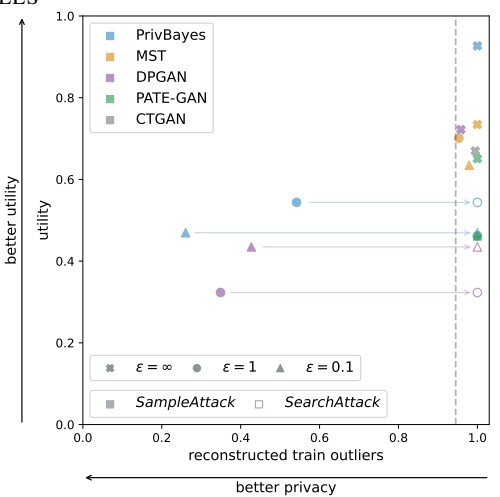

**Figure 6:** Synth data utility vs. proportion of reconstructed train outliers by *ReconSyn*, *Adult Small*.

We report the privacy-utility trade-off in Fig. 6. Regardless of the attacked model, applied privacy budget, or achieved utility, *ReconSyn* is successful at recovering more than 95% of its targets (note the dashed vertical line).

**Utility Evaluation.** Utility is measured through the lenses of similarity, aiming to be consistent with other studies (Tao et al., 2022). More precisely, we report a single similarity score between train and synthetic data by calculating all 1-way marginals and 2-way mutual information scores (normalized between 0 and 1) and averaging them. As expected, applying DP generally reduces utility. Breaking down the effect on the models of the same type, we see that MST's drop is much lower than for PrivBayes. The same occurs for PATE-GAN compared to DPGAN. This is due to the specific DP mechanisms used by the different models, as studied by (Ganev et al., 2023).

**Privacy Evaluation.** Privacy is expressed as the performance of *ReconSyn* in terms of the proportion of reconstructed train outliers. Applying DP to the models with higher utility (MST and PATE-

GAN) does not even defend against *SampleAttack*. Although applying DP does protect against *SampleAttack* for PrivBayes and DPGAN, this comes with a big drop in utility (as discussed above). Nonetheless, *SearchAttack* recovers all train outliers against these two models too.

Even though DP does not help, this does not mean that DP does not work. In fact, in this context, the leakage comes from the privacy metrics; as they require access to the train data and are deterministic (as discussed in Issue 3 in Sec. 3), they break the end-to-end DP pipeline. No matter what other privacy mechanism is added on top of the metrics, it is unlikely to mitigate the problem.

### 5.2.2 Low Utility Generative Models

**Next, we demonstrate that *ReconSyn* is successful even when attacking generative models with severely restricted capabilities, such as Independent and Random, on *Adult Small*.**

Since neither model has the ability to model the data well, the adversary would not be able to locate the clusters with outliers through *OutliersLocator*. Instead, we set their goal to reconstruct *any* train data points. Keeping the same settings, we launch *SampleAttack* for 1,000 rounds.

**Evaluation.** The adversary manages to recover around 79% of the train data against both models. Unsurprisingly, the recovery rate on Random is much slower than Independent, i.e., the adversary needs more rounds to achieve comparable results. Incidentally, in both cases, the adversary reconstructs all 192 train outliers. This could be due to the small data support and randomness component, as both models have a higher chance of generating data points with low probability compared to the five main models, which learn to generate realistic data better.

If the adversary successfully reconstructs a large proportion of the train data points, they could use them to fit *OutliersLocator* and locate the outliers as a last step.

### 5.2.3 Take-Aways

Attacking models trained with DP guarantees (even with $\epsilon = 0.1$) or models with low utility (Independent and Random) does not mitigate *ReconSyn*. In fact, in all cases, the attack manages to reconstruct more than 95% of the train outliers due to access to the privacy metrics.

## 6 Discussion and Conclusion

This paper presents the first in-depth analysis of the most common similarity-based privacy metrics (SBPMs) used in the synthetic data industry. We empirically demonstrate their shortcomings by building *ReconSyn*, a novel reconstruction attack that successfully reconstructs most train outliers.

Our work proves that reasoning about privacy in the context of synthetic data purely through empirical evaluation and SBPMs is inadequate. Worse yet, we show that the privacy metrics/filters commonly used by leading commercial actors are unreliable and inconsistent. The effectiveness of *ReconSyn*, consistently demonstrates that meaningful privacy protections are often inexistent even if all privacy tests pass. In particular, *ReconSyn* is successful even when attacking low-utility generators and models with DP guarantees due to the severe information leakage coming from the access to the metrics. In all cases, we can completely reconstruct and thus single out and link to most outliers, failing two of the required GDPR privacy guarantees. As a result, synthetic data whose privacy is guaranteed through SBPMs cannot be considered anonymous.

Broadly, we can compare providing privacy through SBPMs to the privacy guarantees of the Diffix system, which are often insufficient (Pyrgelis et al., 2018; Gadotti et al., 2019; Cohen & Nissim, 2020a). Even though the functionalities are different – the former return synthetic data and statistical pass/fail tests and scores, the latter answers to queries – both allow for an unlimited number of queries while not implementing robust privacy mechanisms like DP, ultimately leading to severe privacy violations.

We argue that it is crucial for practitioners to prioritize privacy concerns and rely on established notions of privacy from the academic community to avoid potential catastrophic outcomes. (In App. I, we include further discussion on DP and future research directions.)

**Ethics Statement.** Our goal is not to undermine companies' products but to demonstrate how essential it is to emphasize privacy considerations and rely on established academic notions of privacy when deploying real-world systems. Even though we are not directly attacking deployed systems or accessing/processing any personal data, we shared our work with the two main synthetic data companies using SBPMs (Gretel and MOSTLY AI) in the spirit of responsible disclosure. We have provided them with more than 90 days for a response per Google Project Zero's recommendations and offered to keep the paper confidential until submission. As of September 28, 2023, they have responded to our notice, and we are currently working with them on the next steps.

**Reproducibility Statement.** We make considerable efforts to make our work reproducible. First, we clearly state all of our assumptions throughout the paper. Second, we provide references and step-by-step explanations of how we accessed and prepared the datasets and generative models used in our evaluation. Third, we include a detailed description and pseudocode of our new attack. Last, we intend to share the code with the reviewers/ACs during the discussion period and eventually publicly (once the paper is published).

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

| Company | Funding | Compliance | DP | Metrics/Filters |
|---|---|---|---|---|
| Gretel (Gretel, 2023b) | $67.7M | ✓ | ✓ | SF, OF (Gretel, 2021) |
| Tonic (Tonic, 2023b) | $45.0M | ✓ | ✓ | DCR (Tonic, 2023a) |
| MOSTLY AI (MOSTLY AI, 2023b) | $31.1M | ✓ | ✗ | IMS, DCR, NNDR (MOSTLY AI, 2022b) |
| Hazy (Hazy, 2023a) | $14.8M | ✓ | ✓ | DCR (Hazy, 2023b) |
| Syntegra (Syntegra, 2023) | $5.6M | ✓ | ✗ | IMS, DCR (Syntegra, 2021) |
| DataCebo (DataCebo, 2023) | $3.0M | – | ✗ | IMS (DataCebo, 2022) |
| Synthesized (Synthesized, 2023b) | $2.8M | ✓ | ✓ | SF (Synthesized, 2023a) |
| Replica Analytics[1] (Replica, 2023) | $1.0M | ✓ | ✗ | SF (Replica Analytics, 2020) |
| Statice[2] (Statice, 2023b) | – | ✓ | ✓ | DCR[3] (Statice, 2023a) |

[1] Acquired by Aetion in Jan 2022.
[2] Acquired by Anonos in Nov 2022; integrated into Anonos Data Embassy.
[3] Used by linkability and inference risk metrics (Giomi et al., 2023).

**Table 2:** Synthetic data companies, along with funding (as of publicly available information) and whether they claim to be offering fully regulatory-compliant synthetic data as well as the privacy metrics they use.

Santiago Zanella-Béguelin, Lukas Wutschitz, Shruti Tople, Victor Rühle, Andrew Paverd, Olga Ohrimenko, Boris Köpf, and Marc Brockschmidt. Analyzing information leakage of updates to natural language models. In *ACM CCS*, 2020.

Jun Zhang, Graham Cormode, Cecilia M Procopiuc, Divesh Srivastava, and Xiaokui Xiao. Privbayes: private data release via bayesian networks. *ACM TODS*, 2017.

Xinyang Zhang, Shouling Ji, and Ting Wang. Differentially private releasing via deep generative model (technical report). *arXiv:1801.01594*, 2018.

Zhikun Zhang, Tianhao Wang, Jean Honorio, Ninghui Li, Michael Backes, Shibo He, Jiming Chen, and Yang Zhang. PrivSyn: Differentially Private Data Synthesis. In *USENIX Security*, 2021.

Zilong Zhao, Aditya Kunar, Robert Birke, and Lydia Y Chen. Ctab-gan: effective table data synthesizing. In *ACML*, 2021.

Ligeng Zhu, Zhijian Liu, and Song Han. Deep leakage from gradients. *NeurIPS*, 2019.

## A  COMMERCIAL SOLUTIONS FOR SYNTHETIC DATA

In Q2 2023, we conducted a systematic review of the companies in this space, finding that the main players include: Gretel, Tonic, MOSTLY AI, Hazy, Syntegra, DataCebo, Synthesized, Replica Analytics, and Statice. We then looked for publicly available information about their investment funding, whether they claim to produce regulatory-compliant synthetic data, and their approach to privacy – whether they support DP training and what privacy metrics and/or filters they use. Our findings are summarized in Table 2, while the metrics (Identical Match Share (IMS), Distance to Closest Records (DCR), Nearest Neighbor Distance Ratio (NNDR)) and filters (Similarity Filter (SF), Outlier Filter (OF)) are discussed in detail in Sec. 2.

Almost all companies claim their synthetic data products comply with regulations like GDPR, HIPAA, CCPA, etc., even though there are no established standards for providing privacy in the context of synthetic data. However, we observe some emerging trends in their approach – deploying DP and relying on privacy metrics/filters. Companies with funding above $10M, except for one, report adopting DP. Incidentally, it is not surprising that better-funded companies do so, as integrating DP in production requires specialized technical knowledge. Four of the nine companies we reviewed do not support DP but **claim to** guarantee privacy through one or more of the metrics, while two combine it with privacy filters. **For companies relying solely on privacy metrics, providing access to these metrics for each synthetic data sample becomes essential for attempting to demonstrate privacy to end users. We observe that even when companies implement DP during the training of generative models, the privacy metrics and filters still access the sensitive data without perturbation or proper privacy budget accounting.**

## B  RELATED WORK

**(DP) Generative Models.** Several techniques use generative approaches (and DP) to produce synthetic tabular data, including copulas (Li et al., 2014; Patki et al., 2016), graphical models (Zhang et al., 2017; McKenna et al., 2021; Cai et al., 2021), workload/query based (Vietri et al., 2020;

Aydore et al., 2021; Liu et al., 2021), deep generative models such as Variational Autoencoders (VAEs) (Acs et al., 2018; Abay et al., 2018) and Generative Adversarial Networks (GANs) (Xie et al., 2018; Zhang et al., 2018; Jordon et al., 2018; Xu et al., 2019; Frigerio et al., 2019; Long et al., 2021), and so on (Zhang et al., 2021; Ge et al., 2021). As we discuss later in App. C.1, our evaluation focuses on the best-performing models with public and reliable implementations.

**Reconstruction in Databases.** Dinur & Nissim (2003) present the first reconstruction attack where the adversary can theoretically reconstruct records from a database consisting of $n$ entries by sending count queries and solving a linear program. The adversary can make at most $n$ queries, while the answers must be highly accurate. Follow-up studies (Dwork et al., 2007; Dwork & Yekhanin, 2008) generalize and improve on the results by relaxing some of the assumptions and achieving better reconstruction rates. While these attacks are of theoretical nature, they have contributed to the rigorous privacy definition of DP (Dwork et al., 2006). Also, reconstruction attacks on aggregate statistics contributed to the US Census Bureau's deployment of DP for the 2020 Census (Garfinkel et al., 2019). More recently, Dick et al. (2023) reconstruct private records based on aggregate query statistics **and publicly known distributions** while also reliably ranking them. Overall, prior work on databases does not involve training machine learning models as the adversary has access to either (true) aggregate statistics or the ability to query a private database.

**Reconstruction Attacks in Machine Learning.** These are often seen as an extension of attribute inference attacks and are sometimes referred to as model inversion attacks (Fredrikson et al., 2014; 2015; Yeom et al., 2018). Broadly, attribute inference attacks assume (black-box or white-box) access to a trained model and partial knowledge about a data point, while their goal is to infer the missing attribute(s). On the contrary, our attack, *ReconSyn*, has no access to partial records; we discuss attribute and membership inference attacks (Stadler et al., 2022) and how they could be thought of as subcases of *ReconSyn* in App. H.

Model inversion attacks have been presented in a variety of settings. Zhu et al. (2019) demonstrate how an adversary with access to model gradients can efficiently use them to reconstruct train records; the recovery is pixel-wise accurate for images and token-wise matching for texts. In online and federated learning settings, attackers can infer train data points or their labels from inspecting the intermediate gradients during training (Wang et al., 2019; Geiping et al., 2020; Salem et al., 2020; Zanella-Béguelin et al., 2020). This assumes observing the gradient updates of the target model multiple times, whereas we have black-box access to a single trained model.

Train data extraction attacks, which could also be considered reconstruction, have been proposed in the context of (generative) large language models (Carlini et al., 2019b; 2021) and diffusion models (Carlini et al., 2023). These usually assume some auxiliary knowledge (e.g., the presence of the target in the train data or, similarly to attribute inference, a subset of the target's attributes) and query the model multiple times in order to exploit their memorization vulnerability. In other words, they exploit the tendency of large models to memorize and reproduce the train data at generation. Finally, Balle et al. (2022) and Haim et al. (2022) propose reconstruction attacks against discriminative models in which the adversary either has access to all data points but one or to the trained weights and reconstruct the remaining one/several train records. In contrast, our attack does not assume any knowledge about the train data or the model parameters and also works when the model has no memorization capability.

## C  ADDITIONAL BACKGROUND

### C.1  SYNTHETIC DATA AND DP GENERATIVE MODELS

**Synthetic Data.** We focus on synthetic data produced by generative machine learning models. A sample dataset $D_{train}^n \in Z$ (consisting of $n$ iid records drawn from domain $Z$) is used as input to the generative model training algorithm $G(D_{train}^n)$ during the fitting step. Next, $G(D_{train}^n)$ updates its parameters $\theta$ to capture a representation of $P(D_{train}^n)$ and outputs the trained model $G_{\bar{\theta}}(D_{train}^n)$. Then, $G_{\bar{\theta}}(D_{train}^n)$ can be used to repeatedly sample a synthetic dataset (of arbitrary size $n'$) $D_{synth}^{n'} \sim G_{\bar{\theta}}(D_{train}^n)$. Finally, we use $D_{train}^{out} \in D_{train}^n$ to denote the outliers or train records with low density.

**Differential Privacy (DP)** is a mathematical definition that formally bounds the probability of distinguishing whether any given individual's data was included in the input dataset by looking at the output of a computation (e.g., a trained model).

More formally, a randomized algorithm $\mathcal{A}$ satisfies $(\epsilon, \delta)$-DP if, for all possible outputs $S$, and all neighboring datasets $D$ and $D'$ ($D$ and $D'$ are identical except for a single individual's data), it holds that (Dwork et al., 2006; Dwork & Roth, 2014):

$$P[\mathcal{A}(D) \in S] \leq \exp\left(\epsilon\right) \cdot P[\mathcal{A}(D') \in S] + \delta \qquad (1)$$

Note that $\epsilon$ (aka the privacy budget) is a positive, real number quantifying the information leakage, while $\delta$, usually an asymptotically small real number, allows for a probability of failure. DP is generally achieved through noisy/random mechanisms that could be combined together, as the overall privacy budget can be tracked due to DP's *composition* property. Also, through the *post-processing* property, DP-trained models can be re-used without further privacy leakage.

**(DP) Generative Models.** We focus on two types of generative approaches, graphical models (PrivBayes and MST) and GANs (DPGAN, PATE-GAN, and CTGAN), as they are generally considered to perform best in practice **in the tabular domain** (NIST, 2018). All algorithms have open-source implementations and rely on different modeling techniques and, when applicable, DP mechanisms. **We also present two baseline models, Independent and Random.** Except for CTGAN, all support DP training.

Essentially, graphical models break down the joint distribution of the dataset to explicit lower-dimensional marginals, while GANs approximate the distribution implicitly by training two neural networks with opposing goals – a generator, creating realistic synthetic data from noise, and a discriminator, distinguishing real from synthetic data points.

**PrivBayes (Zhang et al., 2017)** follows a two-step process – finding an optimal Bayesian network and estimating the resulting conditional distributions. First, it builds the network by iteratively selecting a node that maximizes the mutual information between the already chosen parent nodes and one of the remaining candidate child nodes. Second, it computes noisy distributions using the Laplace mechanism (Dwork et al., 2006).

**MST (McKenna et al., 2021)** uses a similar approach. First, it utilizes Private-PGM (McKenna et al., 2019) (method inferring data distribution from noisy measurements) to form a maximum spanning tree of the underlying correlation graph by selecting all one-way and subset of two-way marginals (attribute pairs). Then, the marginals are measured privately using the Gaussian mechanism (McSherry & Talwar, 2007).

**DPGAN (Xie et al., 2018)** modifies the GAN training procedure to satisfy DP. It uses DP-SGD (Abadi et al., 2016) to sanitize the discriminator's gradients (by clipping the norm of individual ones and applying the Gaussian mechanism to the sum), which guarantees the privacy of the generator by the post-processing property.

**PATE-GAN (Jordon et al., 2018)** combines an adapted PATE framework (Papernot et al., 2017; 2018) with a GAN to train a generator, $t$ teacher-discriminators, and a student-discriminator. The teacher-discriminators are presented with disjoint partitions of the data and are optimized to minimize their loss with respect to the generator. The student-discriminator is trained on noisily aggregated labels provided by the teacher-discriminators while its gradients are sent to train the generator.

**CTGAN (Xu et al., 2019)** is one of the most widely used non-DP generative models. It uses mode-specific normalization to overcome the non-Gaussian and multimodal distribution of mixed-type tabular datasets. It relies on a conditional generator and training-by-sampling to capture data imbalances.

**Independent & Random** are used as baseline models. The former (Ping et al., 2017; Tao et al., 2022; Mahiou et al., 2022) is commonly used as a baseline for (DP) synthetic data generation. It models all columns independently, thus attempting to preserve the marginal distributions but omitting higher-order correlations. The latter has even lower utility as it randomly samples from the distinct values from all columns.

| Dataset | Cardinality | #Columns | #Records | #Train Outliers |
|---|---|---|---|---|
| *2d Gauss* | $10^5$ | 2 | 2,000 | 108 |
| *25d Gauss* | $10^{31}$ | 25 | 2,000 | |
| *Adult Small* | $10^5$ | 6 | 6,000 | 192 |
| *Adult* | $10^{15}$ | 14 | 6,000 | 116 |
| *Census* | $10^{43}$ | 41 | 10,000 | 193 |
| *MNIST* | $10^{78}$ | 65 | 10,000 | 488 |

**Table 3:** Summary of datasets.

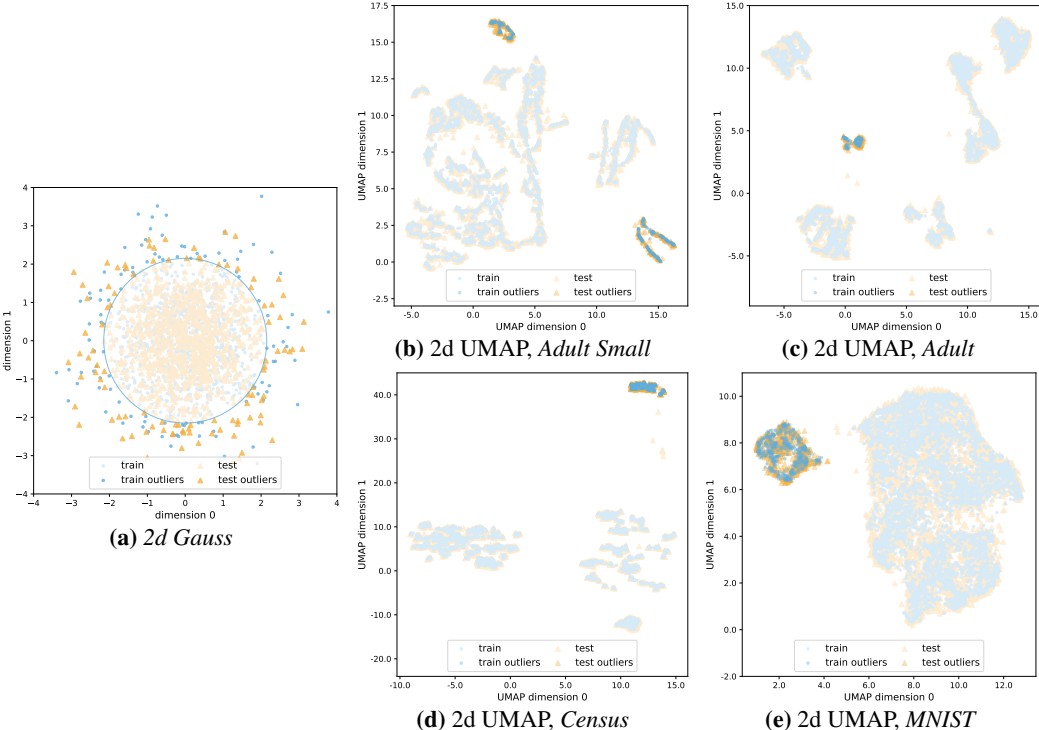

**(a)** *2d Gauss*

**(b)** 2d UMAP, *Adult Small*

**(c)** 2d UMAP, *Adult*

**(d)** 2d UMAP, *Census*

**(e)** 2d UMAP, *MNIST*

**Figure 7:** Train and test data, *2d Gauss* and 2d UMAP reduced, *Adult Small*, *Adult*, *Census*, *MNIST*

## C.2  DATASETS

**We outline the datasets used for our evaluation and our criteria for defining outliers**. A summary of the datasets is provided in Table 3, with visual representations in Fig. 7. We construct two controllable datasets based on the normal distribution and choose three "standard" datasets. Our experiments show that *ReconSyn* can effectively reconstruct train outliers, which are diversely characterized and situated across the datasets.

***2d Gauss*.** We sample 2,000 points from a standard bivariate normal distribution with zero correlation. We consider all train points beyond the blue circle displayed in Fig. 7a (centered at 0, radius 2.15) outliers, or about 10%.

***25d Gauss*.** This is similar to *2d Gauss* but extended to 25 dimensions (again, standard normal distribution). We do not reconstruct outliers for this dataset.

***Adult*.** We use two versions of the Adult dataset (Dua & Graff, 2017). We randomly sample 6,000 data points and refer to this dataset as *Adult*. For *Adult Small*, we further simplify it by selecting six columns ("age," "education," "marital status," "relationship," "sex," and "income"). For both datasets, we fit Gaussian Mixture model with 10 clusters, for the former we select the smallest cluster to be outliers, while for the latter the smallest two (2d UMAP reduction shown in Fig. 7b–7c).

***Census.*** We randomly sample 10,000 data points from the Census dataset (Dua & Graff, 2017). In order to determine the outliers, we fit Gaussian Mixture model with 4 clusters and select the smallest one (2d UMAP reduction is plotted in Fig. 7d).

***MNIST.*** We sample 9,000 data points from the digits "3," "5," "8," and "9" from the *MNIST* (LeCun et al., 2010) dataset as well as 1,000 from "0" and treat them as outliers (2d UMAP reduction displayed in Fig. 7e). In order to simplify the dataset, we downscale the images to 8x8 pixels and discretize all pixels to 16 bins.

**Outliers Definition.** While various definitions of outliers exist in literature (Carlini et al., 2019a; Meeus et al., 2023), we define underrepresented data regions and outliers in a way that captures various scenarios, aiming for an intuitive selection of roughly 10% of the train data (to serve as our targets as noted in Table 3). For the *2d Gauss* dataset, we identify outliers as points lying beyond a certain distance from the center. In the *Adult Small*, *Adult*, and *Census* datasets, outliers are the smallest clusters determined by a Gaussian Mixture model. For *MNIST*, the digit '0' is deliberately underrepresented. We apply the same strategy or fitted model to the test data. UMAP's distance-preserving feature allows these outlier identification strategies to be visually verified in Fig. 7.

## D    FURTHER LIMITATIONS OF SIMILARITY-BASED PRIVACY METRICS

We present further limitations of SBPMs in addition to the ones discussed in Sec. 3.

***Issue 6: Incorrect Interpretation.*** From a statistical theory point of view, the results of the privacy metrics pass/fail tests can be misinterpreted. Assuming we have a good statistical test, the null hypothesis ($H_0$) or the statement we hope not to find enough evidence to reject is "privacy is preserved," while the alternative hypothesis ($H_A$) becomes "privacy is not preserved." When the observed data supports $H_A$, we can reject $H_0$ and claim that we have detected privacy violations. However, when the observed data does not allow us to reject $H_0$, this simply means that we fail to reject $H_0$. We cannot claim that $H_0$ is accepted or "privacy is preserved/guaranteed."

***Issue 7: Risk Underestimation.*** Most implementations of the privacy metrics require discretizing numerical columns and using Hamming distance to compute the similarity between data points. Unfortunately, the calculations become imprecise, and the privacy protections are overstated compared to relying on, for example, continuous data and Euclidean distance.

***Issue 8: Practical Limitations.*** Last but not least, there often are important implementation challenges. Due to the sensitive nature of the data, it is imperative to train the generative model within the secure environment the data resides. Once trained, the model cannot be exported since the privacy metrics require access to the train data for each generation run. This prompts a challenge where accessing the secure environment becomes necessary for every synthesized data.

Furthermore, the metrics need a 50/50% split between train and test data. This could hurt the performance of the model and, consequently, the quality of the synthetic data, particularly when dealing with limited data.

## E    PRIVACY METRICS COUNTER-EXAMPLES

In this section, we present six counter-examples highlighting the untrustworthiness and inconsistency of the privacy metrics/filters. For the first five, we use *2d Gauss* (shown in Fig. 7a). Since all attributes are continuous, we use the Euclidean distance to make the computations more accurate.

***1. Leaking All Test Data.*** Assume a synthetic dataset that is an exact replica of the test data. All privacy tests pass as the two distributions of distances (($D_{train}^n$, $D_{synth}^{n'}$) and ($D_{train}^n$, $D_{test}^n$)) are identical. Following the supposed guarantees provided by the metrics, we would be free to release this dataset. Naturally, publishing half of the sensitive records cannot be considered privacy-preserving.

***2. Leaking All Train Outliers.*** Next, assume that the synthetic data contains all train outliers (with an indistinguishably small perturbation) and the value (0, 0) repeated five times the size of the

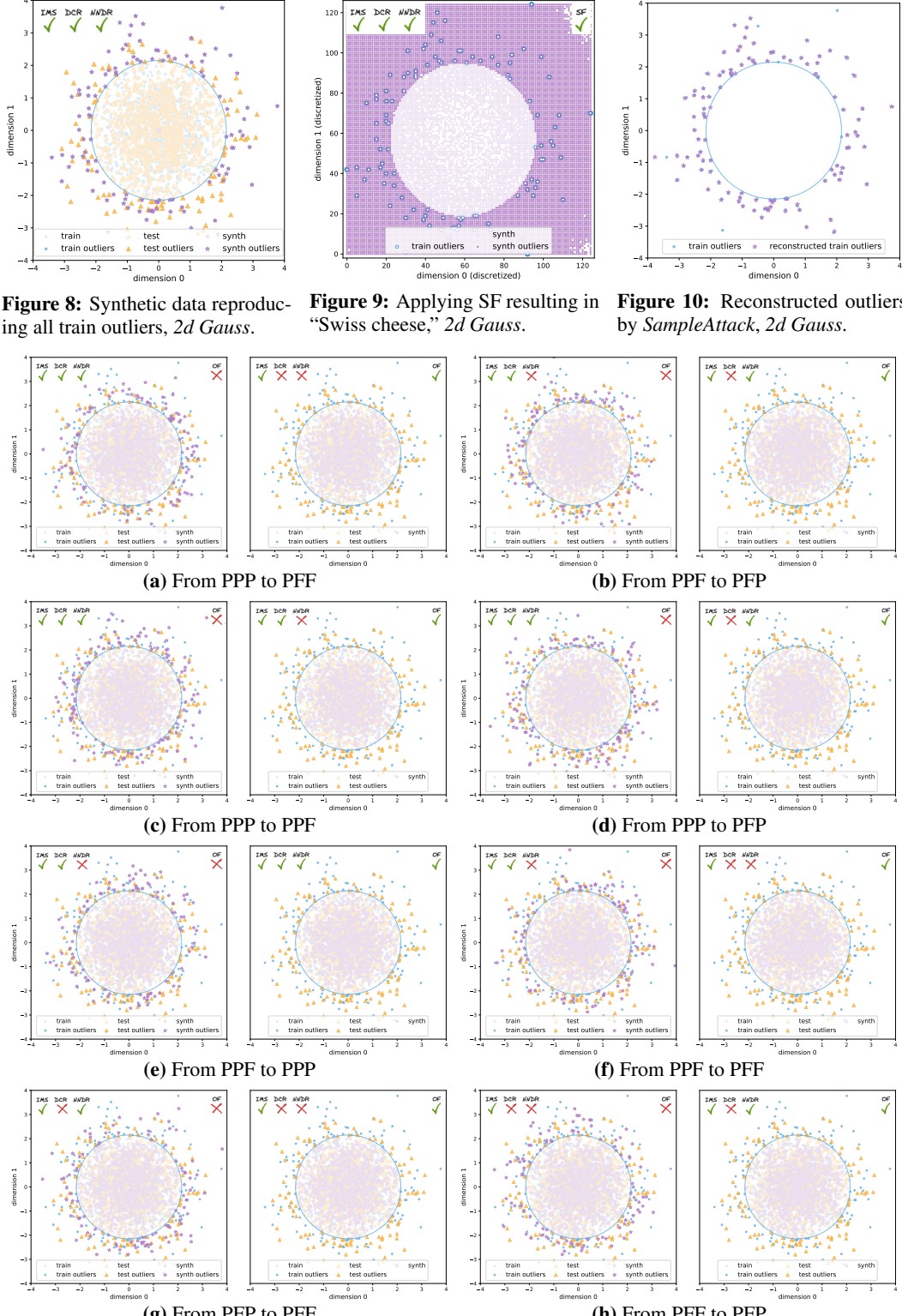

**Figure 8:** Synthetic data reproducing all train outliers, *2d Gauss*.

**Figure 9:** Applying SF resulting in "Swiss cheese," *2d Gauss*.

**Figure 10:** Reconstructed outliers by *SampleAttack*, *2d Gauss*.

**(a)** From PPP to PFF

**(b)** From PPF to PFP

**(c)** From PPP to PPF

**(d)** From PPP to PFP

**(e)** From PPF to PPP

**(f)** From PPF to PFF

**(g)** From PFP to PFF

**(h)** From PFF to PFP

**Figure 11:** Examples of privacy metrics unreliability and inconsistency before and after applying OF, *2d Gauss*.

train data, as displayed in Fig. 8. Again, all privacy tests pass: there are no exact matches, and even though the synthetic outliers are extremely close to the train ones, the large sample size of zeros skews the distances enough to fool both DCR and NNDR. If this synthetic dataset is released, individuals whose data corresponds to the outliers and whose sensitive attributes are leaked would be unconvinced that their privacy is actually preserved (ONS, 2018).

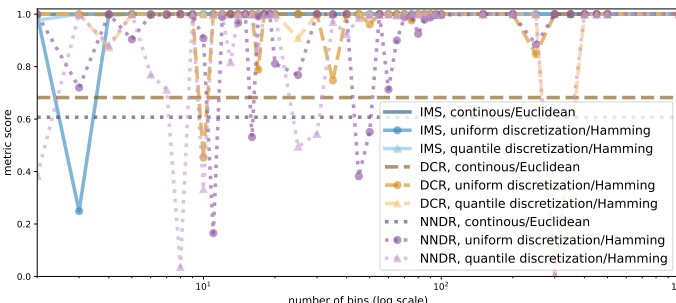

**Figure 12:** Discretization effect on privacy metrics, *25d Gauss*.

*3. Metrics Inconsistency.* We assume access to an oracle possessing knowledge of the train/test data generative process. Suppose we rely on the oracle to sample 1,000 new datasets, act as if they are synthetic datasets, and use them as input to the privacy metrics. Since no generative model was trained (i.e., the train data was never exposed to a model), any data directly sampled from the oracle perfectly preserves the privacy of the train data. A good privacy metric should, therefore, reflect that by reporting a high privacy score.

Only on 274 occasions (out of 1,000) did all the privacy tests pass. This demonstrates that the metrics, and in general empirical approaches measuring privacy of a single synthetic dataset, completely fail to capture the generating process.

Moreover, the proportion of times when the individual metrics IMS, DCR, and NNDR pass is, respectively, 1, 0.48, and 0.38, which is widely inconsistent. Even though the synthetic datasets were sampled from a fixed distribution, which can be thought of as a generator, DCR and NNDR report random results which are not close to 0 or 1. In practice, this means that even if the generative model captures the underlying generating process well, without overfitting or memorizing the train data, the pass/fail decision depends on a specific sample, is noisy, and cannot be trusted.

Alternatively, if we fix a synthetic dataset, randomly split the available sensitive data into 50/50% train/test datasets, and feed them into the metrics, we again run into inconsistencies. Out of 1,000 repetitions, only 380 instances pass all three tests. This highlights the inherent randomness in the train/test split, which incorporates instability into the evaluation process.

*4. OF & Metrics Inconsistency.* We also examine how applying the OF, which is supposed to always improve privacy, affects it according to the metrics. We again rely on the oracle to draw a few synthetic data samples.

On the left plot of Fig. 11a, we see that the synthetic data passes all 3 tests, while on the right plot, when the outliers are filtered out, both DCR and NNDR fail. Even more surprisingly, Fig. 11b shows that removing the outliers can cause a previously failing test to pass (NNDR) and vice versa (DCR). These examples serve as further evidence of the untrustworthiness and inconsistency of the privacy metrics and filters.

Fig. 11c–11h present additional inconsistent results when applying OF. We observe all possible combinations of DCR and NNDR, from passing to failing (in most cases independently from each other) and the other way around.

*5. SF & Reconstruction.* We can reconstruct *all* outliers if SF is applied. Once again, we use the oracle. We sample 100,000 synthetic datasets (we apply SF at generation and select only datasets that pass all privacy tests) and plot them all in Fig. 9.

Clearly, all train outliers can immediately be detected and reconstructed from the "holes" in the data. We refer to this emerging pattern as "Swiss Cheese." This simple experiment shows that, even though SF could naively be considered an additional privacy layer, filtering data out actually *exposes* data points. Furthermore, outliers could uniquely be identified since they are typically in low-density regions. This phenomenon is also discussed in (Jordon et al., 2022), although not demonstrated through any actual experiment, measurement, or visualization.

*6. Discretization Effect.* Finally, in Fig. 12, we measure the effect of discretizing data. We test two discretization strategies, i.e., uniform and quantile while varying the number of bins from 2 to 1,000. For discrete data, we use Hamming distance, while for continuous data, Euclidean. For

| Model | Any Train Records | | Train Outliers |
| | *Sample* | *Search* | *Search* |
|---|---|---|---|
| PrivBayes | 0.20 | 0.98 | (0.95) |
| MST | 0.33 | 0.91 | (0.74) |
| DPGAN | 0.04 | 0.85 | (0.51) |
| PATE-GAN | 0.07 | 0.85 | (0.50) |
| CTGAN | 0.06 | 0.84 | (0.48) |

**Table 4:** Reconstruction of any train records by *ReconSyn*, *Adult*.

this example, we test on *25d Gauss*, use an oracle to sample 1,000 synthetic datasets, and report averages.

The continuous results show that both DCR and NNDR report average values roughly around 0.65 (again failing to capture the generating process, similarly to previous examples). Second, discretizing the data and using Hamming distance greatly overestimates privacy – DCR and NNDR have scores of around 1, except for some randomly looking drops. Incidentally, the metrics report approximately correct results, but for the wrong reasons, i.e., they overestimate privacy due to discretization but fail to capture the generating process. Last, varying the discretization strategy and the number of bins does not help any metrics become more accurate (closer to the continuous baseline).

**Take-Aways.** The privacy metrics/filters appear inconsistent and untrustworthy as one could trick them into labeling clearly non-private scenarios as private.

## F    *ReconSyn* ALGORITHM STEPS

Here, we provide more details about the algorithmic steps of *ReconSyn*, as shown in Algorithm 1.

*OutliersLocator*. We implement two strategies for selecting outliers in order to cover a wide set of scenarios: they could lie outside the cluster(s) (*2d Gauss*) or within the clusters (*Adult Small*, *Adult*, *Census*, and *MNIST*).

*SampleAttack*. As previously mentioned, responses from the metrics API are limited to cases where all three tests pass (line 13). However, our second counter-example in App. E demonstrates that these tests can be easily tricked to expose exact distances. Specifically, by manipulating or augmenting the input synthetic data, we can precisely determine the distance between a target synthetic record and the nearest train data counterpart. One effective method involves submitting the target point alongside about 100 copies of a frequently appearing record. This approach essentially simulates a scenario where the metrics API reveals individual distances to all submitted data (which we adopt for lines 13, 24, and 31), allowing for the detection of matches with 100% confidence.

*SearchAttack*. First, we expand on the definition of record's neighboring dataset and its role in identifying the unreconstructed columns for that record (lines 22–26). The neighboring could be build as follows: we create a square matrix by duplicating the record $d$ times ($d$ is the number of columns), then we alter the values along the diagonal by some amount. When this dataset is fed into the metrics API, the distances to the columns that have been accurately reconstructed will most likely increase, since their values have been moved away from an exact match. Consequently, this indicates that the unchanged/closer columns need to be corrected.

Second, we take a closer look at how we determine the correct values for these columns (lines 27–33). To reduce the number of calls to the metrics API, which could be excessive if all possible combinations were enumerated, we implement a greedy strategy. This involves iterating over the columns one at a time and building all potentially closer candidates by going over the possible values for the current column. Additionally, to further minimize the number of API calls, we use *OutliersLocator* and recorded history to filter out unsuitable candidates.

One could argue that *SampleAttack* and *SearchAttack* are brute-force approaches. Using the Hamming distance greatly limits the efficiency of the adversary, as it does not provide any sense of direction (at all times, any value is either an exact match or not). Therefore, the attack could possibly be improved further; nonetheless, it achieves strong performance in reconstructing the train outliers (see Sec. 5.1) and is computationally practical—for all settings in our experiments, both phases run in less than 24 hours on an m4.4xlarge (16 CPUs, 64GB RAM) AWS instance.

---

**Algorithm 1** *ReconSyn* **Attack**

---

**Require:**
    Trained Generative Model, $G_{\overline{\theta}}$
    Privacy Metrics, $M$ **(namely, IMS, DCR, NNDR)**
    Size of train data, $n_{train}$
    Size of train outliers, $n_{out}$
    SampleAttack rounds, $r_{sma}$
    SearchAttack target distances, $d_{sra}$

1: **procedure** OUTLIERSLOCATOR($G_{\overline{\theta}}, n_{train}, n_{out}$)
2:     Generate $S \leftarrow G_{\overline{\theta}}.\text{sample}(3 \cdot n_{train})$     ▷ **generate new synthetic data**
3:     **Initialize, fit, and predict** $c_{out} \leftarrow GM.\text{fit\_predict}(S)$   ▷ **fit and predict Gaussian Mixture model**
4:     **Select** $c_{out} \leftarrow \max\{c \subseteq c_{out} : \sum_{c' \in c} |c'| \leq n_{out}\}$   ▷ **select clusters containing outliers**
        **return** $c_{out}, GM$
5: **end procedure**
6: **procedure** SAMPLEATTACK($G_{\overline{\theta}}, M, GM, r_{sma}, c_{out}$)
7:     Initialize $R_{sma} \leftarrow \emptyset$     ▷ **initialize SMA reconstructed outliers to the empty set**
8:     Initialize $H_{out} \leftarrow \emptyset$     ▷ **initialize history to the empty set**
9:     **for** $r$ in $r_{sma}$ **do**     ▷ **iterate over number of rounds**
10:         **Generate** $S \leftarrow G_{\overline{\theta}}.\text{sample}(n_{train})$     ▷ **generate new synthetic data**
11:         **Select** $S \leftarrow \{S[i] \mid GM.\text{predict}(S)[i] \in c_{out}\}$     ▷ **select outliers candidates**
12:         **Filter** $S \leftarrow S \setminus H_{out}$     ▷ **filter candidates out from history**
13:         **Query** dists $\leftarrow M(S)$     ▷ **query metrics (augment if necessary)**
14:         **Update** $R_{sma} \leftarrow R_{sma} \cup \{S[i] \mid dists[i] = 0\}$     ▷ **update reconstructed outliers**
15:         **Update** $H_{out} \leftarrow H_{out} \cup \{\text{Zip}(S, dists)\}$     ▷ update history
16:     **end for**
        **return** $R_{sma}, H_{out}$
17: **end procedure**
18: **procedure** SEARCHATTACK($M, GM, d_{sra}, c_{out}, , H_{out}$)
19:     Initialize $R_{sra} \leftarrow \emptyset$     ▷ **initialize SRA reconstructed outliers to the empty set**
20:     **Select and sort** $H'_{out} \leftarrow \{H_{out}[i] \mid dists[i] \leq d_{sra}\}$     ▷ **trim history**
21:     **for** $s, dist_s$ in $H'_{out}$ **do**     ▷ **iterate over history**
22:         **Build** $N_s \leftarrow \{s \text{ with } s[i] \text{ modified}, \forall i \in [1, \text{length}(s)]\}$   ▷ **build record neighboring dataset**
23:         **Filter** $N_s \leftarrow N_s \setminus H_{out}$     ▷ **filter candidates out from history**
24:         **Query** dists $\leftarrow M(N_s)$     ▷ **query metrics (augment if necessary)**
25:         **Update** $H_{out} \leftarrow H_{out} \cup \{\text{Zip}(N_s, dists)\}$     ▷ update history
26:         **Select** $c_s \leftarrow \{i \mid dists[i] \leq dist_s\}$     ▷ **find columns yet to be reconstructed**
27:         **for** $c_{s_i}$ in $c_s$ **do**     ▷ **iterate over columns yet to be reconstructed**
28:             **Build** $CC_s \leftarrow \{s_i \mid \forall val \in \text{Support}(c_{s_i})\}$     ▷ **build column closer candidates**
29:             **Select** $CC_s \leftarrow \{CC_s[j] \mid GM.\text{predict}(CC_s)[j] \in c_{out}\}$   ▷ **select outliers candidates**
30:             **Filter** $CC_s \leftarrow CC_s \setminus H_{out}$     ▷ **filter candidates out from history**
31:             **Query** dists $\leftarrow M(CC_s)$     ▷ **query metrics (augment if necessary)**
32:             **Update** $R_{sra} \leftarrow R_{sra} \cup \{CC_s[j] \mid dists[j] = 0\}$   ▷ **update reconstructed outliers**
33:             **Update** $H_{out} \leftarrow H_{out} \cup \{\text{Zip}(CC_s, dists)\}$     ▷ update history
34:         **end for**
35:     **end for**
        **return** $R_{sra}$
36: **end procedure**

---

## G   RECONSTRUCTION OF ALL TRAIN DATA

**To validate the hypothesis that reconstructing outliers is inherently more difficult (stated in Sec. 4), we conduct an experiment aimed at recovering *any* train records from *Adult*. We impose stricter constraints than those in Sec. 5.1: we limit *SampleAttack* to only 250 rounds, down from 1,000, and allow *SearchAttack* to trace just 3 distances, instead of 4. The results are summarized in Table 4.**

**We observe that even under these limited settings, *ReconSyn* manages to reconstruct a significant proportion of the train data. The performance exceeds this when the goal was to recover only the outliers and was allowed more computations. The outcome should not come as a surprise, given that outliers are less likely to be generated. This is further supported by the last column of the table, which illustrates a substantial drop in recovering the outliers for all models, PrivBayes being the only exception.**

## H    RELATION TO OTHER PRIVACY ATTACKS

*ReconSyn* is a powerful and general reconstruction attack. With similar setups, other attacks like membership and attribute inference could be considered specific subcases of *ReconSyn*, in fact, using considerably less computation. In the rest of this section, we discuss this in more detail, starting with the changes required in the threat model.

**Membership Inference (Shokri et al., 2017; Stadler et al., 2022; Hayes et al., 2019).** In a typical setup, a membership inference adversary has access to a target record, $r_t$ (entire record, i.e., all attributes), representative data, and the model's training algorithm. Usually, they fit several shadow models – i.e., models aiming to mimic the behavior of the model under attack – to infer whether $r_t$ was part of the train data. To adapt *ReconSyn* to this setting, we start with the same assumptions discussed in Sec. 4, i.e., the adversary only has black-box access to a single trained generative model and the privacy metrics. Besides $r_t$, the adversary does not need any other side information (i.e., representative data or the model's training algorithm).

The attack becomes relatively simple: the adversary: i) generates a synthetic dataset, ii) sends two calls to the privacy metrics ($D_{synth}^{n'} \cup r_t$ and $D_{synth}^{n'}$, i.e., one with and one without the target record), and iii) observes the outputs. If the outputs are the same – in particular, if IMS is unchanged – then the target is not a member. Otherwise, the attacker can confidently infer that the target was part of the train data.

**Attribute Inference (Yeom et al., 2018; Stadler et al., 2022).** Here, the adversary has access to a partial target record, $r_t$ (i.e., all attributes but one), which was part of the train data, and the goal is to infer the missing attribute. Adapting *ReconSyn*, the attacker does not need any other side knowledge beyond $r_t$, the trained generator, and privacy metrics.

To mount the attack, the adversary: i) generates a synthetic dataset, ii) sends $k$ calls with different values of the unknown attribute (distinct categories or the number of bins if the attribute needs discretization) to the privacy metrics ($D_{synth}^{n'} \cup r_{t_1}, D_{synth}^{n'} \cup r_{t_2} \ldots D_{synth}^{n'} \cup r_{t_k}$), and iii) observes the outputs. We can now identify the value of the unknown attribute by observing the output whose IMS is higher. It is reasonable to assume that the missing attribute corresponds to a unique data record, as this was true in our empirical evaluations and, in fact, is highly likely, especially in high dimensional data (Rocher et al., 2019).

The attack could be extended to $t$ unknowns attributes, similar to the attack in (Oprisanu et al., 2022). The chance of multiple reconstructed records increases, however, leading to potentially reduced precision.

**Take-Aways.** *ReconSyn* could easily be adapted to accommodate membership and attribute inference scenarios. Assessing a target record's presence in the train data and/or recovering their unknown attribute(s) could be achieved confidently with a handful of computations.

## I    DISCUSSION AND FUTURE WORK

### I.1    REMARKS ON DIFFERENTIAL PRIVACY (DP)

**Benefits of DP.** Our analysis highlights several undesirable weaknesses stemming from SBPMs, which we exploit to build our reconstruction attack. *Training generative models while satisfying DP does address SBPM's drawbacks.* In particular, privacy becomes an attribute of the process (as also advocated in (Trask et al., 2020)); by virtue of DP's post-processing property, any synthetic data sample becomes differentially private too. Also, using DP provably prevents singling out predicates (Cohen & Nissim, 2020b) and empirically decreases singling out, linkability, and inference attacks, which are closely related to the privacy risks outlined in the EU's Article 29 Data Protection Working Party (Giomi et al., 2023). Regulators (Information Commissioner's Office, 2022) and researchers (López & Elbi, 2022; Ganev, 2023) also advise using DP to protect the privacy of outliers.

**Challenges of DP.** However, our review of the product offerings shows that DP is not the standard; alas, companies in this space rarely use it and, in some cases, seem to argue against it (MOSTLY

AI, 2021). This motivates the need for a systematic review and a formal evaluation of the sector's alternative heuristics that are the de facto standard. Our work does so through an analytical review of the metrics, a series of counter-examples, and the instantiation of an attack (*ReconSyn*) with minimal assumptions and high success.

Also note that there also are limitations in using DP with generative models, which may possibly explain why leading companies opt for heuristics instead. First, since there is no one-model-fits-all for all use cases (Jordon et al., 2022), it must be determined whether DP addresses the right threat. For some instances, DP guarantees (assuming a worst-case scenario) could be too conservative, as a practical adversary may not be capable of launching an attack that reaches the theoretical bounds (Nasr et al., 2021). Also, DP does not safeguard broader confidential information beyond privacy: if the dataset contains proprietary secrets, e.g., company-specific terms/names/locations, they may be exposed in the synthetic data combined with additional PETs like anonymization, sanitization, masking, etc. (NHS England, 2021).

Moreover, selecting the optimal combination of the generative model and DP mechanism is challenging, as it depends on factors such as the privacy budget, downstream task complexity, dataset dimensionality, imbalance, and domain (Ganev et al., 2023). Determining the right privacy budget is highly context-specific and not straightforward (Hsu et al., 2014). Additionally, DP often leads to a decrease in utility due to the introduction of noise or randomness, which affects data records disproportionately, particularly outliers (Stadler et al., 2022; Kulynych et al., 2023) and underrepresented classes/subgroups (Bagdasaryan et al., 2019; Ganev et al., 2022). Combining generative models and DP mechanisms, unfortunately, could result in inherently unpredictable synthetic data; that is, it is not clear what signals/trends will be preserved (Stadler et al., 2022), which is a fundamental property of usable privacy mechanisms (EDPS, 2018). Implementing DP in practice and effectively communicating its properties is also non-trivial (Cummings et al., 2021; Houssiau et al., 2022c).

## I.2 FUTURE WORK

*ReconSyn* successfully reconstructs the majority of outliers in various settings, yet, it could still be optimized. In future work, we plan to relax two assumptions: 1) giving the adversary limited access to the generator, perhaps generating only a certain amount of records, and 2) preventing the adversary from augmenting the generated synthetic data.

Also, if the privacy metrics used a continuous distance (e.g., Euclidean) to measure similarity, apart from making the calculations more precise, it would also open other interesting research avenues, such as better search algorithms.

Finally, while our work demonstrates that SBPMs should not be used to guarantee privacy, empirical evaluations and privacy attacks should not be entirely disregarded. They can be valuable tools to detect flaws, errors, or bugs in algorithms and implementations, aiding in model auditing (Jagielski et al., 2020; Tramer et al., 2022; Nasr et al., 2023), and can enhance the interpretability of theoretical privacy protections (Houssiau et al., 2022a;b). We believe that future work could shed more light on their real-world contributions.

