# OpenReview forum: "On the Inadequacy of Similarity-based Privacy Metrics: Reconstruction Attacks against ``Truly Anonymous Synthetic Data''"
_ICLR.cc/2024/Conference — Submitted to ICLR 2024_

### Official Review · Reviewer_82J9 · 2023-10-23

**Soundness:** 4 excellent
**Presentation:** 3 good
**Contribution:** 2 fair
**Rating:** 5
**Confidence:** 5

**Summary:**

In this work authors empirically demonstrate the insufficiency of the existing privacy metrics when applied to synthetically generated data. In addition, authors also propose a novel data reconstruction attack to highlight the vulnerabilities in generative models.

**Strengths:**

The problem authors address is very important and their work is rather timely.

One particular thing I really like in this work is that this is yet another case, where academia identifies a critical issue, where industry does not, except that this time, it is an issue that directly affects industry and cannot be easily ignored, making this work’s conclusions incredibly relevant to the community.

The experimental setting makes sense to me and I agree that the threat model can represent a valid real-world deployment.

**Weaknesses:**

I do have some concerns, however.

While the numerical results that authors demonstrate are very impressive, their choice of datasets is rather limited: in many cases, this is just a limitation/future work area, however, in this work, i argue this is a major flaw. These datasets were commonly known as the relatively ‘trivial’ ML tasks, mainly because of how simple they are. Therefore, reconstructing this data (as shown in works such as [1,2,3,4]) is A) relatively straightforward even with simple inversion attacks and B) is very often due to a very limited reconstruction space that the adversary is present with (i.e. there are many ways to define a valid number in MNIST, for instance). So while these results demonstrate that the metrics are indeed poorly performing on these specific datasets, I am not convinced this is the case for more complex settings, where the reconstruction space is much larger (e.g. even in CIFAR10). Therefore, I would like to see similar results being presented on non-trivial tasks to show that the proposed flaws do affect more realistic settings too.

One fundamental concern of mine related to this is that this attack may require human supervision: from section 5, it reads that even for cases as simple as MNIST, the adversary needs to ‘strategically’ perform some sort of reconstruction guidance (e.g. excluding certain pixels). I would argue that this, in part, violates your threat model and really harms the applicability of your method to real-world imaging tasks. Because in order to know which pixels need removal, the attacker needs to have some sort of prior on the images: sure it is trivial with handwritten digits what pixels to remove, but what about skin pathology imaging? The images are much more identifiable if one knows attributes associated with individuals in the dataset, but also require much more expertise to know how to interpret them and what to keep/remove.

Additionally, while the authors claim that the models they attack are SotA, I am not fully certain that this is indeed the case. It was previously shown that models such as DPGAN, for instance, are not particularly great at generating meaningful datasets when there are outliers which need to be represented (which is, again, poorly captured in datasets such as MNIST).

Some of the issues with the SBPM metrics are valid, but rather inflated. In particular, I agree that issue 4 correctly identifies the issues when adversarial samples are present - but this is also an assumption applicable to almost all ML settings too? Same for issue 5: while I agree that it is often easy to misinterpret what SPBM mean numerically, the same can be said about DP (which given issue 1 is ‘supposed to be’ an alternative to SPBMs as it gives you objective guarantees), which has a privacy parameter often poorly interpreted and contextualised. So yes, this is an issue, but it is not specific to SPBMs and I would go as far as arguing that it is not fully fair to associate these issues with every single SPBM method: in my view, IMS is extremely easy to interpret, and this is precisely why its problematic (i.e. its interpretation is easy, but the score itself is unrepresentative, causing its misuse).

My concrete suggestion here would be to tone down some of the issues (and maybe consider if all of them are really relevant to these specific methods you present) and then use the space that you gained to include more information on how the attack works in the main body. I understand that the algorithm itself is in the appendix, but there is nothing in the main body at all that helps the reader understand how this attack actually functions. All we have is a threat model and assumptions. The reason I find this important is because you are able to leverage these privacy violations primarily because you were able to present a novel data reconstruction attack, which is the core contribution of this work, but you are really not presenting this main contribution at all (except for the appendix).

**Questions:**

Could you also, perhaps, clarify how exactly was DP training performed? From my understanding, construction of privacy filters requires you to derive information from the private data: hence it is unclear to me how this was handled and accounted for in terms of the privacy budget? This might be related to the leakage comes from the privacy metrics; as they require access to the train data and are deterministic: does this mean that your empirical evaluation is done in this manner and hence uses unaccounted private information? If so, you are leaking information, thereby completely invalidating your DP. Could you clarify this in detail, as DP is a large section of your work and your conclusions strongly suggest that DP cannot mitigate your attack. Also: it seemed to me that all of your algorithms except for CT-GAN used DP already, why a separate section?

Additionally: I would like you to present the SearchAttack in more detail: it is unclear what ‘history of records from the first pass’ is.

Overall, I am really on the fence about this work. The conclusions and empirical examples can be incredibly useful for the ML community as a whole, but the results themselves are not particularly representative and the DP section is very unclear (and can potentially be very misleading). I would like the authors to address the issues I outlined above, in which case I will be willing to increase my score.

[1] - Geiping, Jonas, et al. "Inverting gradients-how easy is it to break privacy in federated learning?." Advances in Neural Information Processing Systems 33 (2020): 16937-16947.
[2] - Zhu, Ligeng, Zhijian Liu, and Song Han. "Deep leakage from gradients." Advances in neural information processing systems 32 (2019).
[3] - Usynin, Dmitrii, et al. "Zen and the art of model adaptation: Low-utility-cost attack mitigations in collaborative machine learning." Proc. Priv. Enhancing Technol. 2022.1 (2022): 274-290.
[4] - Zhang, Yuheng, et al. "The secret revealer: Generative model-inversion attacks against deep neural networks." Proceedings of the IEEE/CVF conference on computer vision and pattern recognition. 2020.

---

> ### Author Response · Authors · 2023-11-16
> **Author Response to Reviewer 82J9 (1/2)**
>
> We thank the reviewer for their detailed review and helpful comments. We made several changes (summarized in the common message above). Here, we address individual comments.
>
> ### Weaknesses:
> > These datasets were commonly known as the relatively ‘trivial’ ML tasks, mainly because of how simple they are. Therefore, reconstructing this data (as shown in works such as [1,2,3,4]) is A) relatively straightforward even with simple inversion attacks and B) is very often due to a very limited reconstruction space that the adversary is present with (i.e. there are many ways to define a valid number in MNIST, for instance).
>
> > I would argue that this, in part, violates your threat model and really harms the applicability of your method to real-world imaging tasks.
>
> The primary focus of our paper, in terms of data modality, is tabular data. All companies and privacy metrics studied in App. A offer solutions for tabular data. Additionally, the studied DP generative models (potentially with the exception of DPGAN) in App. C.1 were originally proposed for tabular data. (we added clarifications in Sec. 1 and App. C.1) As such, Adult and Census are datasets used in numerous studies.
>
> A) Unlike [1,2,3,4] which either have white-box access to the target model and/or observe the gradients and/or posses other auxiliary information such as representable publicly available train data, we (in addition to the metrics, which leak privacy) only assume a black-box access to a single trained generative model and no other side information. Therefore, we cannot use any of the proposed techniques which makes the reconstruction challenging.
>
> B) Compared to previous black-box attacks vs synthetic data, we study more datasets, 6 (vs 2 for both (Stadler et al., 2022, Annamalai et al., 2023), with much higher dimensionality, 65 (vs 18/10 for (Stadler et al., 2022, Annamalai et al., 2023). In other words, the reconstruction space is far greater than previous studies. Also in the case of MNIST, our reconstruction accuracy is pixel wise.
>
> > Therefore, I would like to see similar results being presented on non-trivial tasks to show that the proposed flaws do affect more realistic settings too.
>
> As mentioned above, we used datasets of a much higher dimensionality and domain support. As such, we respectfully argue that they are non-trivial and represent realistic settings in the context of tabular synthetic data.
>
> > It was previously shown that models such as DPGAN, for instance, are not particularly great at generating meaningful datasets when there are outliers which need to be represented (which is, again, poorly captured in datasets such as MNIST).
>
> To the best of our knowledge, DPGAN is still widely used for private tabular data generation, given that a variation of it won a NIST competition [5]. We agree that it potentially is not great at generating underrepresented data records (especially if there is a high imbalance and a strict privacy budget is applied, as visible in Fig. 6). Our experiments confirm that since our attack is consistently least effective against DPGAN. However, this could be interpreted that we do not need high utility models to reconstruct records successfully due to the severe metrics leakage.
>
> > I agree that issue 4 correctly identifies the issues when adversarial samples are present - but this is also an assumption applicable to almost all ML settings too?
>
> In many ML settings, such as classification, regression, etc., we usually care about the average-case or how well the model generalizes. For privacy problems, however, we require some level of guarantee and should instead care about the worst-case scenario. In this context, DP is a worst-case guarantee, while the privacy metrics look at the average-case.
>
> > Same for issue 5: while I agree that it is often easy to misinterpret what SPBM mean numerically, the same can be said about DP (which given issue 1 is ‘supposed to be’ an alternative to SPBMs as it gives you objective guarantees), which has a privacy parameter often poorly interpreted and contextualised.
>
> > My concrete suggestion here would be to tone down some of the issues (and maybe consider if all of them are really relevant to these specific methods you present) and then use the space that you gained to include more information on how the attack works in the main body.
>
> We agree that DP is also often misinterpreted (or difficult to interpret) and discuss that in App. I.1. We moved Issue 5 (now Issue 6) to App. D to free some space and left only the most relevant Issues in the main paper. We use this space to provide an overview of the attack in Sec. 4.

---

> ### Author Response · Authors · 2023-11-16
> **Author Response to Reviewer 82J9 (2/2)**
>
> ### Questions:
> > Could you also, perhaps, clarify how exactly was DP training performed? From my understanding, construction of privacy filters requires you to derive information from the private data: hence it is unclear to me how this was handled and accounted for in terms of the privacy budget?
>
> We agree and do state that training the generator while satisfying DP but allowing unperturbed queries to the metrics breaks the end-to-end DP pipeline -- please see Sec. 5.2.1. We rephrased our statements to make this point more clear in the Abstract and Sec. 1. Our motivation for combining DP model training and direct/unperturbed metrics access is to emulate the product offerings of some companies in practice. We also made this point more clear in Sec. 5.2.1 and App. 1.
>
> > Also: it seemed to me that all of your algorithms except for CT-GAN used DP already, why a separate section?
>
> In Sec. 5.1, we attack the DP generative models while keeping $\epsilon=\infty$. We added a clarification in Table 1. In Sec. 5.2.1, we experiment with different epsilon values, namely 1 and 0.1. We keep our experiments with DP and low-utility generative models in a separate subsection because both of them produce synthetic data with reduced utility.
>
> > Additionally: I would like you to present the SearchAttack in more detail: it is unclear what ‘history of records from the first pass’ is.
>
> We provided more details for both $SampleAttack$ and $SearchAttack$ in Sec. 4 and App. F. In particular, we clarified the steps of $SearchAttack$ in Algorithm 1 in App. F. The history is an internal object of the adversary, where they keep track of all the records they fed into the metrics (as illustrated in steps 16, 27, and 35). This way, the adversary can check what records they have already sent to the metrics API and greatly reduce the overall number of calls.
>
> ### References:
> [5] NIST, The Unlinkable Data Challenge, 2018

---

> > ### Author Response · Authors · 2023-11-21
> > **Thank You**
> >
> > Dear Reviewer `82J9`,
> >
> > With the approaching deadline in mind, we are reaching out to kindly check in on our submission. Your insights and feedback have been highly valuable, and we would be grateful for any updates or additional comments you could provide us concerning our revisions and responses.

---

> > > ### Comment · Reviewer_82J9 · 2023-11-22
> > > **Response to authors**
> > >
> > > I would like to thank the authors for their thorough responses and changes to the manuscript they made!
> > >
> > > Some of my concerns remain unaddressed, however: for instance, authors acknowledge that what is claimed to be DP is not, in fact, fully-DP-trained and there is no secure end-to-end pipeline here. I appreciate that you have now reflected this in the text, but I do not agree that this is a valid way to conduct DP experiments.
> > >
> > > I also appreciate your comments on the dataset selection and the attacks that I linked (they were not meant to represent an identical setting, but rather serve as examples of more challenging dataset-model combinations). And while I agree that having access to a gradient for instance relaxes your threat model (from adversarial perspective), the complexity of the task is still much higher than the one presented in MNIST example and I’m not convinced still that these results scale.
> > >
> > > My score, hence, remains unchanged.

---

> > > > ### Author Response · Authors · 2023-11-23
> > > > **Author Follow-up Response to Reviewer 82J9**
> > > >
> > > > We are thankful to the reviewer for engaging with the discussion process and for providing us with further feedback. We provide further clarification which will, hopefully, address any remaining concerns.
> > > >
> > > > > what is claimed to be DP is not, in fact, fully-DP-trained… I do not agree that this is a valid way to conduct DP experiments.
> > > >
> > > > We completely agree that our experiment in Sec. 5.2.1 does not employ a fully end-to-end DP pipeline, a point we have explicitly clarified in the Abstract, Sec. 1, and Sec. 5.2.1. This intentional design choice was made to critically evaluate the practices of some companies, as outlined in App. 1, which combine DP training of generative models with direct, unperturbed access to privacy metrics. We believe these practices pose significant privacy risks, and our experiment aims to highlight these vulnerabilities. This aligns with our paper's main goal of assessing and demonstrating the limitations of existing privacy metrics.
> > > >
> > > > Furthermore, recent studies [6] on machine learning systems further reinforce our point by demonstrating that side information (in our case privacy metrics/filters, in their case data filters applied before/after training) can compromise the integrity of DP training and leak more privacy even if they are meant to improve privacy.
> > > >
> > > > > I’m not convinced still that these results scale.
> > > >
> > > > We appreciate the reviewer's concern about the scalability. However, we would like to emphasize again that our study is on _tabular_ data. We believe that our research actually represents an advancement in the study of privacy attacks against tabular synthetic data, aligning closely with and extending beyond the scope of prior works (Stadler et al., 2022, Houssiau et al., 2022a;b, Annamalai et al., 2023) and [7, 8, 9]. In these studies, the authors evaluate their proposed (membership/attribute inference) privacy attacks vs at most 2 datasets with dimensionality no more than 35. The Adult dataset is the most commonly used one (which we use as well). In contrast, we run a more challenging task (reconstruction attack) on more datasets, 6. Moreover, some of the datasets included in our evaluation are higher dimensional – 41 (Census) and 65 (MNIST).
> > > >
> > > > Furthermore, our choice of datasets and their dimensionality aligns with evaluation benchmarks studies on state-of-the-art DP generative models (majority of which are not deep learning approaches) for tabular data (Tao et al., 2022). The largest dataset they use is Census, which we use as well. Additionally, some of the top performing DP models on tabular data such as MST (McKenna et al., 2021) do not practically scale beyond 100 columns (Ganev et al., 2023); this further supports our choice of datasets as both challenging and representative of current real-world applications in DP for tabular data.
> > > >
> > > >
> > > > ### References:
> > > > [6] Debenedetti et al., Privacy Side Channels in Machine Learning Systems, arXiv:2309.05610, 2023
> > > >
> > > > [7] Breugel et al., Membership Inference Attacks against Synthetic Data through Overfitting
> > > > Detection, AISTATS, 2023
> > > >
> > > > [8] Meeus et al., Achilles' Heels: Vulnerable Record Identification in Synthetic Data Publishing, arXiv:2306.10308, 2023
> > > >
> > > > [9] Guépin et al., Synthetic is all you need: removing the auxiliary data assumption for membership inference attacks against synthetic data, arXiv:2307.01701, 2023

---

### Official Review · Reviewer_WhWB · 2023-10-29

**Soundness:** 2 fair
**Presentation:** 3 good
**Contribution:** 2 fair
**Rating:** 3
**Confidence:** 4

**Summary:**

The use of synthetic datasets generated from real datasets is common in industrial applications. One of its main applications is to comply with privacy regulations when real datasets are sensitive and cannot be used directly. To assess their privacy before its release, a synthetic dataset is evaluated it by the use of privacy metrics. The main metrics that are used in practice are Identical Match Share (IMS), Distance to Closest Records (DCR) and Nearest Neighbor Distance Ratio (NNDR). However, they evaluate privacy via heuristics that do not provide strong guarantees.

The contribution show that IMS, DCR and NNDR are not good metrics for privacy. They do so by (i) exposing major flaws in the design with respect to standard privacy and security guidelines  (ii) illustrating contradictions in the privacy assessment of the metrics by the use of counter examples and (iii) providing a data reconstruction attack that is able to recover a substantial amount of dataset outliers by having black-box access to the synthetic data generators and the output of the privacy metrics.

**Strengths:**

The paper clearly presents and treats an important problem in the study of privacy preserving applications. It is key to understand the privacy threats posed by the extensive use of synthetic data.

Section 3 and Appendix E show fundamental flaws in the use of IMS, DCR and NNDR as meaningful metrics. Section 3 outlines key aspects of the design of privacy preserving mechanisms which are not present these metrics. Appendix E provides counter examples that illustrate how substantial sensitive data can be revealed even if privacy metrics give a good score to the released data.

**Weaknesses:**

As said, early sections of the paper successfully identify important design flaws in the assessment of the privacy provided by synthetic data. However, the arguments provided by the main contribution, which is the ReconSyn the attack, are not sufficiently solid.

The main objective of the paper is to show that IMS, DCR and NNDR are inadequate as privacy metrics. This essentially means that these metrics fail to assess how much sensitive information is leaked by sharing a synthetic dataset. To do that, the contribution presents an attack that attempts to recover the original data points from the synthetic data generator. However, the attack proposed by the protocol (ReconSyn) seems to have a major flaw: it success does not rely on the sensitive information released by synthetic data, which is only used marginally. The attack heavily relies on the ability to query privacy metrics an arbitrary amount of times. These queries provide a *huge* amount of information: the distance to real data points (given by DCR and NNDR) and the amount of synthetic data points that match the real ones (given by IMS). It is not surprising that one could successfully reconstruct outlier data points with arbitrary access to that information.

The above is confirmed from the results of Section 5.2 which show that the attack performs well regardless of the quality of the synthetic data and even when data is generated with strong DP guarantees (see Section 5.2.3).

My impression over the main take-away of the paper is that revealing the privacy metrics is dangerous. Therefore, the impact of the contribution seems low: it does not provide a compelling argument against the release of synthetic data itself when its IMS, DCR and NNDR scores are not revealed (even if the synthetic data has been evaluated with these metrics).

In addition, I do not think that the evaluation of the attack against differentially private techniques of generation is well addressed. In the current setting, the attacker has the possibility to access information that is not differentially private (i.e the privacy metrics) and has unlimited queries to the privacy mechanism. It is well known that epsilon and delta DP parameters degrade after each query to the mechanism.

**Questions:**

Please elaborate on the weaknesses outlined above.

---

> ### Author Response · Authors · 2023-11-16
> **Author Response to Reviewer WhWB (1/2)**
>
> We thank the reviewer for their time and comments. We made several changes (summarized in the common message above). Here, we address individual comments.
>
> ### Weaknesses:
> > These queries provide a huge amount of information: the distance to real data points (given by DCR and NNDR) and the amount of synthetic data points that match the real ones (given by IMS).
>
> > My impression over the main take-away of the paper is that revealing the privacy metrics is dangerous.
>
> > It is not surprising that one could successfully reconstruct outlier data points with arbitrary access to that information.
>
> This is indeed the main contribution of our paper – through our analysis, counter-examples, and a new privacy attack, we formally show that relying on IMS, DCR, and NNDR to demonstrate/guarantee privacy leaks privacy. This is an inadequate approach and, as the reviewer pointed out, dangerous. Moreover, to the best of our knowledge, we are the first to rigorously study and scrutinize these metrics.
>
> While it might not be surprising to (most) privacy experts, this approach and, in particular, these privacy metrics are adopted by all top companies in the space, as discussed in App. A. In fact, the privacy tests are used to guarantee privacy of cancer patients in a technical report by the EU (Hradec et al., 2022). Furthermore, we would also like to highlight that there are numerous research papers and whitepapers by companies like AWS that rely entirely on these/similar metrics to demonstrate the privacy of their models (we referenced several such studies in Sec. 1 and Sec. 2).
>
> > the contribution presents an attack that attempts to recover the original data points from the synthetic data generator.
>
> We would like to highlight that we attack a single trained synthetic data generator in a black-box manner without any other auxiliary data, such as the training mechanism or representable training data. Whereas, previous studies on synthetic data (Stadler et al., 2022, Annamalai et al., 2023) require many such instances and access to the training mechanism and representable data.
>
> > The attack heavily relies on the ability to query privacy metrics an arbitrary amount of times.
>
> This is a common assumption in privacy attacks vs synthetic data (Stadler et al., 2022, Annamalai et al., 2023), where the adversary can train an arbitrary amount of models. Other papers on generative models (Hilprecht et al., 2019) require sampling 1,000,000 times from the train and test data, while papers on DP auditing [1] require training 100,000 models on a fixed dataset. Finally, unlimited querying to a model/dataset is, for better or worse, common in production settings -- like for the Diffix system (Pyrgelis et al., 2018; Gadotti et al., 2019; Cohen & Nissim, 2020a), as discussed in Sec. 6.
>
> > it success does not rely on the sensitive information released by synthetic data, which is only used marginally.
>
> To address this specific issue, we designed $SampleAttack$ to rely only on synthetic data samples generated by the generative model, i.e., it cannot search for nearby data records. In this sense, the synthetic data quality is central to the success of the attack.
>
> > it does not provide a compelling argument against the release of synthetic data itself when its IMS, DCR and NNDR scores are not revealed.
>
> First, to remove any confusion, we rephrased the introduction of the attack to mention that it is specifically designed to expose the weaknesses of the metrics (IMS, DCR and NNDR) -- please see Sec. 1.
>
> Second, for many providers, similarity-based metrics are the only way they demonstrate/guarantee the privacy of the product to their end users, which makes access to the metrics crucial. We clarified this in App. A. As confirmed by our direct experience with these companies (details had to be omitted to preserve double-blind submission), in commercial settings, restricting access to these metrics would severely limit the desirability of these products.
>
> To provide a practical example, consider the following scenario: a company holds some sensitive data that needs to be shared internally to another team in a private way. The data owner team trains the generative model on the synthetic data platform and gives access to the other team. An adversary that is part of this team could then generate data using the platform. Since every generated synthetic dataset needs to be private, and privacy is guaranteed through the metrics, the metrics need to accompany every generated dataset. The adversary can then run our attack.

---

> > ### Author Response · Authors · 2023-11-16
> > **Author Response to Reviewer WhWB (2/2)**
> >
> > > I do not think that the evaluation of the attack against differentially private techniques of generation is well addressed.
> >
> > > the attack performs well regardless of the quality of the synthetic data and even when data is generated with strong DP guarantees.
> >
> > We agree and do state that training the generator while satisfying DP but allowing unperturbed queries to the metrics breaks the end-to-end DP pipeline -- please see Sec. 5.2.1. We rephrased our statements to make this point more clear in the Abstract and Sec. 1. Our motivation for combining DP model training and direct/unperturbed metrics access is to emulate the product offerings of some companies in practice. We also made this point more clear in Sec. 5.2.1 and App. 1.
> >
> > This experiment confirms the severe leakage coming from the privacy metrics. Even when attacking state-of-the-art DP generative models and low utility generative models (Random and Independent) $ReconSyn$ still manages to reconstruct the majority of the outliers as exemplified in Sec. 5.2.
> >
> > ### References:
> > [1] Tramer et al, Debugging Differential Privacy: A Case Study for Privacy Auditing, arXiv:2202.12219, 2022

---

> > > ### Author Response · Authors · 2023-11-21
> > > **Thank You**
> > >
> > > Dear Reviewer `WhWB`,
> > >
> > > With the approaching deadline in mind, we are reaching out to kindly check in on our submission. Your insights and feedback have been highly valuable, and we would be grateful for any updates or additional comments you could provide us concerning our revisions and responses.

---

> ### Comment · Reviewer_WhWB · 2023-11-22
> **Response to rebuttal**
>
> Dear authors,
>
> I thank you for your detailed responses. I comment on them below.
>
> 1- **Attacks that rely on an arbitrary number of queries**: I do not agree with the authors that existent attacks allow the same kind of queries to sensitive data than this submission:
>
> 1a- It is not true that the attacks proposed in (Stadler et al., 2022, Annamalai et al., 2023) are able to arbitrarily query sensitive datasets. What they are able to do is to train an attacker form a reference dataset or partial non-sensitive data, but they have limited access the sensitive synthetic datasets. I find this substantially different than your attack from a security point of view.
>
> 1b- In [1], the purpose of querying the sensitive dataset many times is done to provide statistically significant estimations of privacy parameters and not to incrementally strengthen the attack as in your case. Therefore the fact that "100,000" models are trained is different and not comparable to your setting.
>
> 1c- Indeed I agree that attacks to the Diffix system that you mention are allowed to arbitrarily query the sensitive information. However, these attacks are realistic because the Diffix system allows any user to do these kind of queries to sensitive data. Therefore, demonstrating that it is not private using the same interface of any user is more relevant. In your contribution, you are attacking a data+metrics generator that, in practice, does not accept arbitrary queries over the sensitive data with your own (manipulated) synthetic dataset. Therefore this does not seem realistic to me.
>
> In addition to the above points, these papers that the authors referenced in their rebuttal have clearly defined the attacker model,  distinguishing games and reconstruction algorithms. Contrary to that, your paper presents its main contribution in the appendix and in a confusing way (see my point below).
>
> 2- **The success of the attack does not rely on the sensitive information released by the synthetic data**: Indeed the loose description of $SampleAttack$ in Section 4 explains that the algorithm can reconstruct information without manipulating the input data points of the metrics. However, the (still loose) explanation of $SampleAttack$ in Appendix F remarks that *"by manipulating or augmenting the input synthetic data, we can precisely determine the distance between a target synthetic record and the nearest train data counterpart."*. Therefore this gives me the impresion that manipulation still occurs and I am not convinced that this is realistic in the current setting. In addition, $SampleAttack$ does not provide the same reconstruction accuracy as the main claims of the paper, which are obtained with $SearchAttack$.
>
>
> 3- **Revealing the metrics is "the only way they demonstrate/guarantee the privacy of the product to their end users"**. I disagree with the authors in this point. Revealing privacy metrics is by no means a proof that these metrics have been computed correctly. An adversary can generate unsafe synthetic datasets and publish them along with fake acceptable metrics. Auditing such behavior is equally difficult regardless of the disclosure or lack of disclosure of the metrics.
>
> Given the points above, my score is likely to remain unchanged.
>
> Refs.
>
> [1] Tramer et al, Debugging Differential Privacy: A Case Study for Privacy Auditing, arXiv:2202.12219, 2022

---

> ### Author Response · Authors · 2023-11-23
> **Author Follow-up Response to Reviewer WhWB (1/2)**
>
> We are thankful to the reviewer for engaging with the discussion process and for providing us with further feedback. We provide further clarification below.
>
> > 1- Attacks that rely on an arbitrary number of queries
> > In your contribution, you are attacking a data+metrics generator that, in practice, does not accept arbitrary queries over the sensitive data with your own (manipulated) synthetic dataset. Therefore this does not seem realistic to me.
>
> We appreciate the reviewer's concern about the realism of attacks relying on arbitrary queries. As clarified in our Ethics Statement, we explicitly state that we are not directly attacking deployed systems but rather a simulated one. Instead, our main contribution is to formally show that relying on IMS, DCR, and NNDR (which are the most commonly used privacy metrics in industry and adopted by numerous research papers) to demonstrate/guarantee privacy is not safe and actually leaks privacy. We do so through our analysis, counter-examples, and a novel proof of concept reconstruction attack.
>
> The core of our argument, detailed in Sec. 1, Sec. 2, and Sec. 4, challenges and disproves the prevailing statement that passing these three privacy tests equates to creating “truly anonymous synthetic data” (Platzer & Reutterer, 2021; Mobey Forum, 2022). Indeed, the goal of the adversary is to “build a collection of synthetic datasets considered private by the provider” which could be used to reconstruct the train outliers. Theoretically, how the adversary builds this collection of synthetic datasets should not be of great importance as long as the individual datasets are deemed safe by the tests. Additionally, the tests, as presented and used in practice, are completely agnostic to the adversary behavior (the do not even provide an adversary) and do not limit the number of queries made to them (infinite amount of synthetic data can be generated). They only take into account the distances between real/test/synthetic datasets (we explain this in Sec. 3 and App. D).
>
> From all potential strategies that could be employed to construct such a collection of datasets, we present one through our attack, $ReconSyn$. The adversary relies on key 3 assumptions, which are all clearly stated in the main paper in Sec. 4. It just so happens that there are companies that not only make our assumptions realistic in practice but also advertise them widely (in short, unlimited data generation, data augmentation, metrics evaluation per every dataset -- we urge the reviewer to carefully go through our citations in Plausibility of the Attack. in Sec. 4).
>
> To reinforce the point that our attack is not a general reconstruction attack but one designed to specifically demonstrate the unreliability of the 3 privacy metrics, me introduce our attack as a proof of concept attack in Sec. 1.

---

> > ### Author Response · Authors · 2023-11-23
> > **Author Follow-up Response to Reviewer WhWB (2/2)**
> >
> > > 2- The success of the attack does not rely on the sensitive information released by the synthetic data
> > > your paper presents its main contribution in the appendix and in a confusing way
> >
> > In response to the helpful reviews and comments by all reviewers, we made significant revisions to address these concerns and improve the presentation of our attack. In Sec. 4 of the main paper, we now provide a comprehensive overview of our attack, including the threat model, key assumptions, motivation, and implications, ensuring that the main elements are clear and accessible. Detailed technical aspects of the attack, due to space constraints, are described in Algorithm 1 in App. F.
> >
> > We wish to clarify that the $SampleAttack$ phase, which may or may not involve data augmentation, is used exclusively to calculate distances from synthetic records generated by the target model to the real data. This augmentation does not manipulate the synthetic records or introduce new sensitive information. Therefore, the success of our attack hinges on the synthetic data released by the model, utilizing discrepancies between real and synthetic data to expose privacy vulnerabilities.
> >
> > To further clarify the mechanics of our attack, we provide a detailed example (Example 2 in App. E) that illustrates how exact distances can be obtained. It illustrates how we can construct a dataset (to be added to our collection mentioned above) that passes all three tests yet still leaks precise distances. We also would like to highlight that data augmentation, as long as the data passes the tests, is a practice advertised by companies, making it a realistic and relevant assumption.
> >
> > > 3- Revealing the metrics is "the only way they demonstrate/guarantee the privacy of the product to their end users"
> >
> > What we tried to convey in our rebuttal comment was that many companies rely exclusively on privacy metrics to evaluate the privacy for a given synthetic dataset as they do not employ DP. Since these companies claim that their products are regulatory-compliant, the only way they can “prove” this to their clients is by sharing the metrics. Besides explicitly advertising it, the companies are often de-facto required by their users to “demonstrate” privacy protection by running, and sharing, the metrics. We use this as a motivation for our attack (which assumes access to the metrics for every generated dataset) rather than agreeing that they actually provide any meaningful notion of privacy.
> >
> > Please note that the adversary cannot manipulate the metrics, they can only make a call to the provider through the metrics API and observe the responses (if they pass) on the platform. We assume that the metrics are always computed correctly. Also, the adversary cannot impersonate, manipulate, or poison the behavior of the provider. We are sorry for any confusion caused.

---

> > > ### Comment · Reviewer_WhWB · 2023-11-23
> > >
> > > Dear authors,
> > >
> > > My assessment/score has been established in my previous comment and remains unchanged. I truly appreciate your responses and the effort on the discussion.

---

### Official Review · Reviewer_XpV4 · 2023-10-30

**Soundness:** 3 good
**Presentation:** 3 good
**Contribution:** 4 excellent
**Rating:** 6
**Confidence:** 3

**Summary:**

This paper explores problems with privacy metrics used to evaluate synthetic data (generated without differential privacy). The paper demonstrates the undesirable properties of these metrics and shows the viability of a reconstruction attack. The paper suggests a need to move away from ad hoc approaches to privacy . Extensive experiments are conducted demonstrating the viability of the attack on real synthetic data under a reasonable threat model.

**Strengths:**

The paper is well motivated, the false promise of synthetic data generation (without DP) is a very important message. The overview of existing metrics for evaluating synthetic data alongside their weaknesses is a very clearly presented and a valuable contribution in its own right.

 The experiments are thorough and extensive detailed is given in the appendix.

Overall the conclusion of the paper that effective attacks on synthetic data exist seems well supported by the experimental results, assuming the presentation can be improved.

**Weaknesses:**

A thorough description of the attack is lacking in section 4 which significantly weakens the comprehensiveness of the paper in my view. I understand why the full Algorithm was relegated to the appendix but a paragraph explain the logic of the attack and where the information is being extracted would significantly improve the paper. Given the vulnerability of models trained with DP to this attack, the information is presumably coming from the metrics but an intuitive explanation of how that occurs is missing.

Figure 2 should be presented in a way where it is possible to distinguish between the different training algorithms and datasets. I would suggest enlarging the figure despite the constrain on space since it is a key result for the paper. I find it more useful that FIgure 4, which is given much more space.

I do not entirely understand what the takeaway is from Figure 3. The figure is referred to only in passing and never explained. More generally
the description of the results would benefit from more thorough explanation.

As far as I can tell, 'outliers' is never clearly defined but it is the denominator for all success rates reported and so it is important to explain what this set of data points is and how much of the training data they make up.

Stating that the attack works on models trained with DP in the abstract/intro is true in a literal sense but I feel somewhat disingenous since as noted later in the paper, the information is come from the statistics released without DP. Any model regarded as formally satisfying DP would never be able to look directly at the private data in this way.

Overall, I believe the paper has a strong attack but these results could be better presented to really hammer the point to ensure the key message of this paper reaches the synthetic data community. Ideally there could be a figure that made it impossible to ignore the weaknesses of the synthetic data generation but the current figures are overly information loaded causing the key takeaway to be lost.

**Questions:**

Can you report the privacy metrics with DP or would your recommendation be to not report distance metrics?

How do you define an outlier? As far as I can tell, 'outliers' is never clearly defined but it is the denominator for all success rates reported and so it is important to explain what this set of data points is and how much of the training data they make up.

---

> ### Author Response · Authors · 2023-11-16
> **Author Response to Reviewer XpV4**
>
> We thank the reviewer for their time and helpful comments and suggestions. We made several changes (summarized in the common message above). Here, we address individual comments.
>
> ### Weaknesses:
> > A thorough description of the attack is lacking in section 4.. a paragraph explain the logic of the attack and where the information is being extracted would significantly improve the paper.
>
> We added an extended description of the attack in Sec. 4 and provided more details in App. F. Furthermore, we updated Algorithm 1 to make it cleaner/easier to follow.
>
> > Figure 2 should be presented in a way where it is possible to distinguish between the different training algorithms and datasets.
>
> We tried to improve the presentation of Fig. 2 (now Fig. 3). Basically, the top 5 lines are from Adult Small, the bottom 5 from Adult, while the middle 5 are from Census. We also reduced the space allocated to Fig. 4 (now Fig. 5).
>
> > I do not entirely understand what the takeaway is from Figure 3.
>
> The main takeaway is that all models generate MNIST digits that are further away from the train data compared to the distance between test data and train data (min distance is 6). This is the main reason $SampleAttack$ performs so poorly (0% for all models). We explain this in Sec. 5.1.1.
>
> > 'outliers' is never clearly defined
>
> We added more details about how we define outliers for every dataset in App. C.2. While there are various definitions of outliers (Carlini et al., 2019a; Meeus et al., 2023), we opt for a simple and intuitive selection of roughly 10% of the train data as outliers, aiming to cover various scenarios. We also explain better how the adversary fits the $OutlierDetector$ and decides on the outliers regions in App. F.
>
> > Any model regarded as formally satisfying DP would never be able to look directly at the private data in this way.
>
> We agree and do state that training the generator while satisfying DP but allowing unperturbed queries to the metrics breaks the end-to-end DP pipeline -- please see Sec. 5.2.1. We rephrased our statements to make this point more clear in the Abstract and Sec. 1. Our motivation for combining DP model training and direct/unperturbed metrics access is to emulate the product offerings of some companies in practice. We also made this point more clear in Sec. 5.2.1 and App. 1.
>
> > these results could be better presented to really hammer the point to ensure the key message of this paper reaches the synthetic data community.
>
> We introduced Fig. 1, which hopefully manages to demonstrate the success of our attack as well as the weakness of the privacy metrics.
>
> ### Questions:
> > Can you report the privacy metrics with DP or would your recommendation be to not report distance metrics?
>
> We are not sure this will scale as querying the metrics in a DP way would eventually deplete the DP budget. This means that no more synthetic datasets would be able to be generated, which contradicts one of the main selling points of synthetic data providers – unlimited querying of the model (generation of new samples of synthetic data).

---

> > ### Author Response · Authors · 2023-11-21
> > **Thank You**
> >
> > Dear Reviewer `XpV4`,
> >
> > With the approaching deadline in mind, we are reaching out to kindly check in on our submission. Your insights and feedback have been highly valuable, and we would be grateful for any updates or additional comments you could provide us concerning our revisions and responses.

---

### Official Review · Reviewer_uGem · 2023-10-31

**Soundness:** 4 excellent
**Presentation:** 2 fair
**Contribution:** 3 good
**Rating:** 8
**Confidence:** 4

**Summary:**

This paper investigates the use of similarity-based metrics as a privacy tool.
Recently, certain companies have developed heuristics to determine when a synthetic dataset is private.
These heuristics have no theoretical backing and are, ironically, a huge privacy leak as they contain deterministic information about the inclusion or exclusion of training samples.
The paper points out a number of issues with these approaches as well as various ways to fool the metrics/inconsistencies in them.
They then develop two variants of an attack that can reconstruct the training data with access to these privacy metrics.
They show that outliers can be reconstructed with high accuracy, and thus, these metrics do not ensure data releases that satisfy GDPR.

**Strengths:**

- It is very important to show that these attempts at privacy-preserving synthetic data do not work (even if obvious to experts) as there are companies using these approaches. I particularly like how it shows an explicit GDPR violation, which will make companies take this more seriously.
- The list of issues is well-written and contains many good points.

**Weaknesses:**

## Paper Structure/ Understanding ReconSyn
Without reading the appendix, I had little to no idea about how the attack works. Then I read the appendix, and I found it still to be rather unclear (although I get the general idea).
1) The attack needs to be described in the main paper. One suggestion to make room for this could be to cut the incredibly verbose evaluation section down to summarize the key points over all datasets with a few interesting anomalies. As the paper is currently written, we read the evaluation with little idea of what is being evaluated.
2) The algorithm needs to be described more formally as to the exact set operation that takes place during filtering and the exact branching conditions based on the metrics.

## How practical is the attack?
While I was convinced that the similarity metrics are a bad idea without the attack, I think the attack is not as practical as it is portrayed.
1) Assuming the adversary can query any dataset with the metric seems far-fetched. Although the paper argues this is the case in practice, it would be easy for companies to restrict this.
2) In Section 5, it seemed the attack was adaptive (The search attack is deployed based on the performance of the Sample attack). However it was not clear how the attack could know the reconstruction rate of the previous component without access to the training data.
3) It would be helpful to list the number of queries to the metric oracles/the effect that a rate-limiting defence would have on attack success.
4) While the attack is useful for a proof of concept, it doesn't have any merit in the case that the privacy metrics are not leaked. Thus, the argument that it outperforms related work in terms of its lack of assumptions is a little misleading.

## DP argument
The section about the effect of DP is somewhat a moot point as it can not be DP while allowing unperturbed queries about the training dataset (the metrics). I somewhat get the idea to show that even if they decided to use DP on the generator, it still wouldn't be enough with the metrics being present. Perhaps more interesting would be to show what happens if the metrics are released with DP (an actual defence). Although epsilon already gives an indicator of how private the synthetic data is, the best defence would be removing the metrics entirely.

## Minor comments
- Consider using parenthesis around citations, as some sentences are hard to parse when filled with citations that are not distinguished from normal text.
- 2D gauss was missing from Table 1 despite section 5.1.1 referring to it.
- No mention of how the ground truth outliers are computed in the evaluation.
- Figure 5 is very hard to read. The shading is subtle and changes the colour to make it even more confusing. Perhaps shapes for eachepsilon and make them shaded or hollow for the two attacks?
- $1/n$ is too large for a delta parameter. It technically allows a trivial mechanism that publishes a single record of the database. The general rule of thumb is that $\delta << 1/n$ [Dwork & Roth](https://www.nowpublishers.com/article/Details/TCS-042)

**Questions:**

I think most of my concerns could be addressed by clarifying the attack and its purpose as a proof of concept (not a general reconstruction attack).

---

> ### Author Response · Authors · 2023-11-16
> **Author Response to Reviewer uGem (1/2)**
>
> We thank the reviewer for the detailed and helpful comments and suggestions. We made several changes (summarized in the common message above). Here, we address individual comments.
>
> ### Weaknesses:
> ## Paper Structure/Understanding $ReconSyn$
> > The attack needs to be described in the main paper.
>
> In the revised paper, we add an overview of the main algorithmic steps describing how the attack works in the main paper in Sec. 4.
>
> > The algorithm needs to be described more formally.
>
> We added more details of the algorithmic steps and clarified Algorithm 1 in App. F.
>
> ## How practical is the attack?
> > Assuming the adversary can query any dataset with the metric seems far-fetched.
>
> First, we would like to clarify that generating unlimited synthetic data from a trained generator is essentially a universal selling point of synthetic data providers. Second, for many providers, similarity-based metrics are the only way they demonstrate/guarantee the privacy of the product to their end users, which makes access to the metrics crucial. We clarified this in App. A. As confirmed by our direct experience with these companies (details had to be omitted to preserve double-blind submission), in commercial settings, restricting access to these metrics would severely limit the desirability of these products.
>
> We would also like to highlight that there are numerous research papers and whitepapers by companies like AWS that rely entirely on these/similar metrics to demonstrate the privacy of their models (we reference several such studies in Sec. 1 and Sec. 2). Finally, unlimited querying to a model/dataset is, for better or worse, common in production settings -- like for the Diffix system (Pyrgelis et al., 2018; Gadotti et al., 2019; Cohen & Nissim, 2020a), as discussed in Sec. 6.
>
> > However it was not clear how the attack could know the reconstruction rate of the previous component without access to the training data.
>
> We now added more details in Sec. 4 and App. F. As discussed in Example 2 in App. E and explained in App. F, we can use $SampleAttack$ to trick the metrics to output the exact distance of a given synthetic record to its closest train record. Thus, we can detect exact matches in the first phase of the attack through the metrics without having access to the train data.
>
> > It would be helpful to list the number of queries to the metric oracles/the effect that a rate-limiting defence would have on attack success.
>
> The number of queries to the metrics for $SampleAttack$ are provided in Sec. 5.1.1; for Small Adult, Adult, and Census, we plot them in Fig. 3. Namely, $SampleAttack$ makes between 1,000 and 5,000 rounds (or calls to the metrics) depending on the dataset. One round involves generating a synthetic dataset and making one call to the metrics API. For comparison, previous papers on generative models (Hilprecht et al., 2019) require sampling 1,000,000 times from the train and test data, while papers on DP auditing [1] require training 100,000 models on a fixed dataset.
>
> Also, as mentioned above, limiting the rates would contradict one of the main advertised reasons for adopting synthetic data – unlimited samples of “private” synthetic data, where privacy is guaranteed through the metrics.
>
> > While the attack is useful for a proof of concept, it doesn't have any merit in the case that the privacy metrics are not leaked.
>
> We rephrased the introduction of the attack to mention that it is a proof of concept attack designed to specifically expose the weaknesses of the metrics in Sec. 1.
>
> ## DP argument
> > It can not be DP while allowing unperturbed queries about the training dataset (the metrics).
>
> We agree and do state that training the generator while satisfying DP but allowing unperturbed queries to the metrics breaks the end-to-end DP pipeline -- please see Sec. 5.2.1. We rephrased our statements to make this point more clear in the Abstract and Sec. 1. Our motivation for combining DP model training and direct/unperturbed metrics access is to emulate the product offerings of some companies in practice. We also made this point more clear in Sec. 5.2.1 and App. 1.
>
> > Perhaps more interesting would be to show what happens if the metrics are released with DP (an actual defence).
>
> We are not sure this will scale as querying the metrics in a DP way would eventually deplete the DP budget. This means that no more synthetic datasets would be able to be generated, which, as mentioned, contradicts one of the main selling points of synthetic data providers.
>
> > Best defence would be removing the metrics entirely.
>
> In principle, we agree. Indeed, the best defense would be to train the model in a DP way and not accompany data releases with metrics. However, as mentioned earlier, for many companies (and research papers), these metrics are the only way to demonstrate/guarantee privacy to customers and users and that is why in practice they are released in the first place.

---

> > ### Author Response · Authors · 2023-11-16
> > **Author Response to Reviewer uGem (2/2)**
> >
> > ## Minor comments
> > > Consider using parenthesis around citations
> >
> > We updated all citations.
> >
> > > Table 1
> >
> > We included 2d Gauss in Table 1.
> >
> > > how the ground truth outliers are computed
> >
> > We added more details about how outliers are defined for every dataset in Sec. 2 and App. C.2.
> >
> > > Figure 5 is very hard to read.
> >
> > We followed the advice and updated Fig. 5.
> >
> > > 1/n is too large for a delta parameter.
> >
> > While we agree, we selected it to follow the default hyperparameter from (Jordon et al., 2018). For the purposes of our evaluation, however, we do not believe the changing delta to $1/n^2$ would make any difference.
> >
> > ### References:
> > [1] Tramer et al, Debugging Differential Privacy: A Case Study for Privacy Auditing, ​​arXiv:2202.12219, 2022

---

> > ### Comment · Reviewer_uGem · 2023-11-17
> > **Response to the Authors**
> >
> > I would like to thank the authors for their detailed responses to my concerns. I have two follow-up concerns:
> >
> > 1) While I appreciate the authors adding some additional high-level descriptions of the attack, The lack of precise algorithm details is still a concern of mine (and appears to be shared by most reviewers). The algorithm needs to be described more rigorously, defining the exact operation taking place on each set of records in each step. For example (not an exhaustive list):
> > - Algorithm 1 was not made more precise; only the comments were updated.
> > - The $\gets$ operator seems very overloaded. Sometimes, it is an assignment; other times, it is filtering (which still needs to be defined).
> > - I think line 12 should be $S\backslash{}H$. i.e. removing any records of S that are in H?
> > - It also needs to be made clear when the $\gets$ operator refers to adding to the set vs. replacing the set. e.g. Line 10 is a replacement, and 16 is adding to the history?
> > - The new text describing the sample attack mentions adding 100 additional copies of a frequent record. I am having trouble finding that in Algorithm 1?
> > - How is the neighbourhood built in Line 24?
> >
> >
> > 2) I think the paper makes it clear that these metrics should not be used. The paper does a solid job of showing they are not a good measure of privacy and actually decrease privacy. However, the authors seem to argue in the rebuttal that making them DP is not a good option (which is fair, given the large number of queries). But they also argue these metrics are the only way to demonstrate privacy for most companies. I think this contradicts the paper. I think a very clear stance needs to be taken to ensure that industry does not continue to allow these blatant privacy violations.

---

> > > ### Author Response · Authors · 2023-11-19
> > > **Author Follow-up Response to Reviewer uGem**
> > >
> > > We are thankful to the reviewer for engaging with the discussion process and for providing us with further feedback. We uploaded a revised manuscript with the following updates:
> > >
> > > 1. We made Algorithm 1 more precise and added further clarification in App. F.
> > >     * We updated most lines of the algorithm, now they describe the exact operations.
> > >    * Correct, thanks for the pointer for line 12, we have fixed that.
> > >    * Correct, line 10 is assigning/replacement, since we are getting the output of a function, while line 16 (now 15) is updating, since the operation is the union of two sets.
> > >    * To keep the notation clear, we explain in App. F that we can follow the second counter example in App. E and get exact distances for any target synthetic records by augmenting the input (while passing all tests). We added a sentence to clarify that we adopt this strategy in App. F.
> > >    * We added more information in App. F (and the operation in line 28). In short, for a given record, the neighboring dataset is a square matrix formed by duplicating the record d times (where d is the number of columns in the record). This matrix is constructed by slightly altering the values along its diagonal. Consequently, each row in this matrix matches the original record, except for one distinct value. Thus, by making a call to the metrics with this dataset, we can track changes in the distances. This enables us to identify which columns in the record require further reconstruction.
> > >
> > > 2. We agree and indeed, the paper (in particular, see Abstract, Sec. 1, and Sec. 6) and rebuttal tries to argue that the similarity-based privacy metrics should not be used by practitioners (and researchers) to “demonstrate/guarantee” privacy of their products. We hope our message is clear and consistent -- these metrics are an inadequate mechanism to reason about privacy in the context of synthetic data and practitioners should not deviate from well established notions of privacy from the academic community.
> > >
> > >     What we tried to convey in our rebuttal comment (last question in `DP argument`) was that many companies rely exclusively on privacy metrics to evaluate the privacy for a given synthetic dataset as they do not employ DP. Since these companies claim that their products are regulatory-compliant, the only way they can "prove" this to their clients is by sharing the metrics. Besides explicitly advertising it, the companies are often de-facto required by their users to “demonstrate” privacy protection by running, and sharing, the metrics. We use this as a motivation for our attack (which assumes access to the metrics for every dataset) rather than agreeing that they actually provide any meaningful notion of privacy. We changed App. A, to make this more clear. We are sorry for any confusion caused.

---

> > > > ### Author Response · Authors · 2023-11-21
> > > > **Thank You**
> > > >
> > > > Dear Reviewer `uGem`,
> > > >
> > > > With the approaching deadline in mind, we are reaching out to kindly check in on our submission. Your insights and feedback have been highly valuable, and we would be grateful for any updates or additional comments you could provide us concerning our revisions and responses.

---

> > > > > ### Comment · Reviewer_uGem · 2023-11-22
> > > > >
> > > > > I would like to thank the authors for a very detailed and productive rebuttal. My concerns have been addressed and I have increased my score to reflect this.

---

> > > > > > ### Author Response · Authors · 2023-11-23
> > > > > > **Thank You**
> > > > > >
> > > > > > We would like to express our gratitude to the reviewer for all the time, patience, insights, and advice shared, which have ultimately improved our paper.

---

### Official Review · Reviewer_5urh · 2023-10-31

**Soundness:** 3 good
**Presentation:** 3 good
**Contribution:** 3 good
**Rating:** 8
**Confidence:** 4

**Summary:**

This paper challenges the effectiveness of similarity-based privacy metrics currently used by start-ups to validate the privacy guarantees of synthetic data. The authors provide a thorough review of these metrics and provide counter-examples where a metric would declare synthetic data to be safe even though it isn’t. The authors then propose a threat model for attacking synthetic data as typically provided by a start-up, where the adversary has unlimited query access to the generative model and the ability to recompute the privacy metrics over arbitrary datasets. They develop a reconstruction attack and demonstrate on several datasets that the attack succeeds in reconstructing a large fraction of outliers in the datasets. Finally, they show that even when the generative model is DP, reconstruction still succeeds better than expected due to leakage in the privacy metrics.

**Strengths:**

- 1) The question being studied is timely and of great practical importance. There are many start-ups in the synthetic data space making broad claims about the privacy of synthetic data and there is not enough scientific scrutiny of these claims.
- 2) Solid review of the privacy metrics used to validate the privacy of synthetic data.
- 3) Provides counter-examples for all of the metrics where the metric would declare the synthetic data to be safe even though it isn’t.
- 4) Proposes and evaluates the first record-level reconstruction attack against synthetic data.
- 5) The attack results are very strong as a large fraction of outliers can be correctly reconstructed.

**Weaknesses:**

- 1) The attack results are contingent on the definition of outliers. Both the targets of the attack (outliers) and the test records (over which the attack success rate is computed) are constructed using similar clustering methods. This may lead to over-optimistic results. Have the authors considered (1) evaluating their attack on a dataset that has not been pre-processed by them to add outliers or (2) identifying outliers (to be used as the test records) with a state-of-the-art approach such as https://arxiv.org/abs/2306.10308?
- 2) It’s unclear which of the similarity-based metrics is used in the empirical evaluation.
- 3) Missing method description in the main paper and insufficient method description in the Appendix. In the main paper, the authors should clearly explain the difference between SampleAttack and SearchAttack (without which interpreting the results is very difficult). In the Appendix, to ensure reproducibility, the authors should precisely describe steps 18-32 of Alg. 1. The current textual descriptions are too informal/imprecise.

Minor (suggestions for improvement):

- 4) Missing (conceptual and/or empirical) comparison to the reconstruction method of Dick et al. Can the authors describe (add a paragraph on) how their reconstruction method relates to Dick et al.? Even though the methods operate on different objects (aggregates in Dick et al. as compared to synthetic data here), they both involve repeated generation of datasets of “candidate” records and filtering of these records in some way to only keep the most likely records.
- 5) Unfinished sentence on page 3: The test passes if [...]-5th percentile is larger or equal.”

**Questions:**

- 1) Is it possible, using the authors' method, to assign a confidence or likelihood to each reconstructed record?
- 2) Why do the authors focus on outliers only? Intuitively, an average record is more likely to be generated by the synthetic data model than an outlier record. The other leading reconstruction method based on repeated dataset generation, developed by Dick et al. against aggregate statistics, doesn’t explicitly target outliers.
- 3) What do the authors mean by “all three tests pass” (relating to the metrics)?

---

> ### Author Response · Authors · 2023-11-16
> **Author Response to Reviewer 5urh**
>
> We thank the reviewer for the encouraging and thoughtful comments. We made several changes (summarized in the common message above). Here, we address individual comments.
>
> ### Weaknesses:
> > Why do the authors focus on outliers only? Intuitively, an average record is more likely to be generated by the synthetic data model than an outlier record.
>
> We agree that the average record is more likely to be generated and thus probably easier to reconstruct. To confirm this hypothesis, we run new experiments, which we discuss in App. G. In short, reconstructing any train records is indeed much easier.
>
> However, in the main body of the paper, we focus on outliers because they likely correspond to the most vulnerable individuals, they are more challenging to attack (which puts our attack to the test), are more susceptible to memorization, and are specifically highlighted by regulators (we also discuss this in Sec. 4). Therefore, if we manage to reconstruct them successfully, this will constitute a clear and serious privacy violation.
>
> > The attack results are contingent on the definition of outliers. Both the targets of the attack and the test records are constructed using similar clustering methods. This may lead to over-optimistic results.
>
> We add more details about how we define outliers for every dataset in App C.2 (including new references to (Carlini et al., 2019a; Meeus et al., 2023)). While the adversary follows the same steps as the provider to determine the outliers regions, we would like to clarify two points: 1) the targets of the attack are the outliers from the train data, not the test data; 2) there is no leakage from the provider to the adversary, i.e., the adversary fits $OutlierDetector$ on their own sample of synthetic data (using the fitted generative model) without having access to any of the train/test data. Therefore, if the adversary does not do a good job with $OutlierDetector$, they will reduce the attack’s performance. All these, also combined with our experiments in Sec. 5.2.2 and App. G (in which the adversary reconstructs large proportions of any train data), lead us to believe that our results could actually serve as a lower bound on the number of records that could potentially be reconstructed.
>
> > It’s unclear which of the similarity-based metrics is used in the empirical evaluation.
>
> The adversary in Sec. 5 has access to the 3 privacy metrics (IMS, DCR, NNDR). In the revised paper, we highlight this in Sec. 2 and Algorithm 1 in App. F. We do so to emulate the real-world product offerings of some companies.
>
> > Missing method description in the main paper and insufficient method description in the Appendix. In the Appendix, to ensure reproducibility, the authors should precisely describe steps 18-32 of Alg. 1.
>
> In the revised paper, we provide a description of the two phases of the attack in the main paper in Sec. 4. Furthermore, we add more details of the algorithmic steps and clarified Algorithm 1 in App. F.
>
> > Can the authors describe (add a paragraph on) how their reconstruction method relates to Dick et al.?
>
> To initiate their successful attack, Dick et al. (2023) need access to the publicly known data distribution and true answers to a number of queries. They use this information to generate multiple synthetic datasets using a RAP variant (Aydore et al., 2021), ranking them based on the number of duplicated rows. In contrast, our attack does not require such any information, except for the privacy metrics. Additionally, our method has a black-box access to a single generative model which does not depend on preserving the actual statistics of the data (which is often the case for the models we evaluate).
>
> > Unfinished sentence on page 3
>
> We fix the sentence.
>
> ### Questions:
> > Is it possible, using the authors' method, to assign a confidence or likelihood to each reconstructed record?
>
> Since the metrics are deterministic and unperturbed (as discussed in Sec. 3), the adversary can be 100% confident for each reconstructed record. We tried to allude to that in the take-aways in Sec. 4.
>
> > What do the authors mean by “all three tests pass” (relating to the metrics)?
>
> Every metric is associated with a statistical test. The passing criteria are either the average or 5th percentile comparison between ($D_{train}, D_{test}$) and ($D_{train}, D_{synth}$). We made this more clear in Sec. 2.

---

> > ### Author Response · Authors · 2023-11-21
> > **Thank You**
> >
> > Dear Reviewer `5urh`,
> >
> > With the approaching deadline in mind, we are reaching out to kindly check in on our submission. Your insights and feedback have been highly valuable, and we would be grateful for any updates or additional comments you could provide us concerning our revisions and responses.

---

### Author Response · Authors · 2023-11-16
**Summary of Changes**

Summary of changes grouped by topics and reviewers who requested them. All changes are reported in orange in the revised manuscript.

$ReconSyn$ Attack:
* Reframed our attack as specifically designed to exploit the privacy metrics rather than a general reconstruction attack in Sec. 1 (`uGem`, `WhWB`).
* Added a description of the attack in Sec. 4 (`5urh`, `XpV4`, `82J9`).
* Clarified the algorithmic steps in Algorithm 1 and added further details in App. F (`5urh`, `XpV4`, `82J9`).

Outliers definition:
* Expanded on the definition of outliers and added motivation for outliers selection in App. C.2 (`uGem`, `XpV4`).
* Added a pointer to the description of outliers in the last paragraph in Sec. 2 (`uGem`, `XpV4`).
* Highlighted the increased difficulty of modeling and reconstructing outliers in Sec. 4 (`5urh`, `XpV4`, `WhWB`).
* Ran a new experiment demonstrating that reconstructing any train data is an easier task than only focusing on outliers in App. G (`5urh`).
* Explained how the adversary uses $OutliersDetector$ to detect outliers for every dataset in App. F (`5urh`).

DP experiments:
* Reiterated and clarified that integrating DP generative model training with unperturbed access to privacy metrics/filters compromises the overall DP framework in Abstract and Sec. 1 (`uGem`, `XpV4`, `WhWB`, `82J9`).
* Provided more detailed settings and the rationale for the DP experiments (we follow what companies do in practice) in Sec 5.2.1 and App. A (`uGem`, `XpV4`, `WhWB`, `82J9`).

Other:
* Added brackets in citations (`uGem`).
* Specified that the generative models are state-of-the-art for tabular data in Sec 1 and App C.1 (`82J9`).
* Moved Issue 5 (now Issue 6) to App. D (`82J9`).
* Added results for 2d Gauss in Table 1 (`uGem`).
* Introduced a new figure (Fig. 1) aiming to demonstrate the success of ReconSyn  (`XpV4`).
* Updated Fig. 2 (now Fig. 3) (`XpV4`).
* Reduced the size of Fig. 4 (now Fig. 5) (`XpV4`).
* Made Fig. 5 (now Fig. 6) more readable (different shapes for each epsilon and full/hollow for the $SampleAttack$/$SearchAttack$) (`uGem`).

---

### Meta-Review · Area_Chair_roNb · 2023-12-12

**Metareview:**

This paper discusses several heuristic "similarity-based" metrics that have been recently proposed to evaluate the privacy of synthetically generated data. It first describes a number of high-level issues with such heuristic measures. It then designs a reconstruction attack that uses multiple releases of such privacy metrics to reconstruct parts of the training dataset. As pointed out by several reviewers the paper addresses a very important question of whether heuristic notions of privacy protect privacy. However the attack itself relies on the value of the metric being released repeatedly for data generated by the same model and not the leakage from a single release of synthetic data and the metric. This is an important distinction as it requires a significantly stronger threat model. One can easily limit the number of times a privacy metric is released for synthetic data generated from a given model on a given sensitive dataset via other means in most practical scenarios. Notably, repeated release of metrics violates privacy even when the generative model is produced with differential privacy. Thus I find the claim that this work establishes inadequacy of these privacy metrics as rather misleading. It does establish an issue with releasing the metrics but not the metric itself. In particular, one can easily prevent the attack by privatizing the metric without actually changing the synthetic data.  While the updated version of the paper addresses and qualifies some of the specific misleading points I think that this work needs to be reframed and rewritten to avoid this major misrepresentation.

**Justification For Why Not Higher Score:**

N/A

**Justification For Why Not Lower Score:**

N/A

---

### Decision · Program_Chairs · 2024-01-16

Reject